# LEARNING CONTINUALLY AT PEAK PERFORMANCE WITH CONTINUOUS CONTINUAL BACKPROPAGATION

## ABSTRACT

Training neural networks under non-stationary data distributions, as in continual supervised and reinforcement learning, is hindered by loss of plasticity and representation collapse. While recent approaches employ periodic, full neuron reinitialization to sustain gradient flow and restore plasticity, they often sacrifice performance and still suffer frequent collapse. To address these limitations, we propose *Continuous* Continual Backpropagation (CCBP), which instead continuously, partially resets units. Empirically, CCBP outperforms both decay-based and reset-based methods after long sequences of distribution shifts, and uniquely prevents policy collapse in challenging continual reinforcement learning environments. Ablations further show how CCBP can be tuned to smoothly trade off plasticity and performance, highlighting gradual reinitialization as a promising direction for continual deep learning.

## 1 INTRODUCTION

Deep neural networks have been shown to excel in modeling arbitrarily complex modalities, leading to great successes in difficult problems such as protein folding (Abramson et al., 2024), language modeling (Brown et al., 2020; Kaplan et al., 2020), image synthesis (Rombach et al., 2022), game-playing (Schrittwieser et al., 2020; Vinyals et al., 2019) and robotics (Zitkovich et al., 2023). However, they appear to struggle when trained continually on non-stationary data: they lose *plasticity* and subsequently their ability to learn from new data is reduced (Dohare et al., 2021; Lyle et al., 2022; 2023; Dohare et al., 2024). This phenomenon presents a crucial obstacle to overcome in order to, for instance, extend the knowledge cutoff of pre-trained large models, or more importantly, unlock the ability for reinforcement learning (RL) agents to learn more effectively from longer training horizons and adapt to changing environments.

Existing work has attempted to elucidate the mechanism through which this loss of plasticity might occur. Lyle et al. (2022) highlight that continual learning can lead to decreasing feature rank, reduced capacity and representation collapse, while Sokar et al. (2023) we observe that shifts in the training distribution cause an increase in previously active neurons becoming dormant. Dohare et al. (2021; 2024) additionally show a decrease in stable rank to correlate with lessened adaptation ability and Lyle et al. (2023) posit that unstable loss landscapes and optimization difficulties can contribute to plasticity loss.

A recent and promising line of work aims to remedy these symptoms by adaptively reinitializing units that are deemed low-utility during training, effectively restoring network capacity. These methods differ primarily in the statistics they base their utility score on, as well as in the way they measure said statistics. Continual Backpropagation (CBP) (Dohare et al., 2021; 2024) defines a unit's utility as the running average of its activation weighed by its outgoing weights, which it also zeros for reinitialization. ReDo (Sokar et al., 2023) computes a simpler activation-based score on batches of data and ReGraMa (Liu et al., 2025) proposes to use gradient magnitude instead to better align with learning capacity in modern architectures. However, these methods fully reset a unit if its utility score is below an arbitrary threshold, which can cause training instability, dampening the effects of increased plasticity and making them susceptible to representation collapse.

To address this limitation and stabilize training while recycling dormant neurons, we propose to avoid the brittleness of binary thresholds and use periodic *partial resets*: rather than fully refresh units, pull their parameters back toward their initialization, and instead of zeroing their outgoing

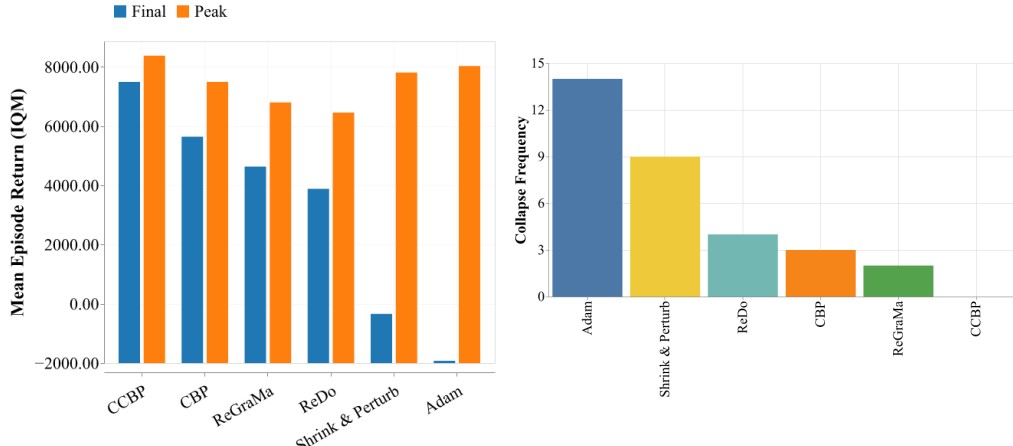

Figure 1: **Left** - Peak vs. final interquartile mean (IQM) episodic return in SlipperyAnt (higher is better). **Right** - number of seeds where policy collapses during training, with 15 seeds total (see subsection 5.4). CCBP achieves peak return comparable to classical optimizers without observed collapses.

weights, shrink them. Additionally, both of these operations can be applied to *all* units, similar to decay-based approaches for stabilizing gradient updates (Ash & Adams, 2019), but with the reset and dampening amount modulated by a per-unit utility score.

We demonstrate the benefits of this approach, which we name *Continuous* Continual Backpropagation (CCBP), on two non-stationary reinforcement learning control tasks from MuJoCo (Slippery-Ant and Slippery-Humanoid), where the friction of the environment changes periodically. Due to the increased training stability, CCBP maintains comparable plasticity levels to existing methods, while reaching and maintaining higher task performance and reducing policy collapse events. Our contributions can be summarized as follows:

- We demonstrate that current reset-based continual learning algorithms suffer from unstable training dynamics
- We develop a novel method (CCBP) to stabilize continual learning by replacing binary threshold reinitialization with continuous network unit resets
- We conduct experiments on MuJoCo tasks with added non-stationarity, showing that the increased training stability in CCBP allows it to reliably regain peak performance after environment changes and avoid policy collapse
- We ablate max per neuron reset $\rho$ for CCBP and demonstrate that it can be used to trade off plasticity and post-adaptation performance

## 2 BACKGROUND

### 2.1 PROBLEM SETTING

In the reinforcement learning problem, the environment can be formalized by a Markov Decision Process (MDP), defined by the tuple $\mathcal{M} = (\mathcal{S}, \mathcal{A}, \mathcal{P}, \mathcal{R}, \gamma)$, where $\mathcal{S}$ denotes the state-space, $\mathcal{A}$ the action-space, $\mathcal{P}$ the transition probability function $\mathcal{S} \times \mathcal{A} \times \mathcal{S} \rightarrow [0, 1]$, $\mathcal{R}$ is the reward function $\mathcal{S} \times \mathcal{A} \rightarrow \mathbb{R}$ and $\gamma \in [0, 1)$ is the discount factor. The agent interacting with the environment follows a potentially stochastic policy $\pi : \mathcal{S} \times \mathcal{A} \rightarrow [0, 1]$. The goal of the agent is to learn the optimal policy $\pi^*$ which maximizes the episodic return $\mathbb{E}_{a_t \sim \pi^*}[\sum_{t=0}^{\infty} \gamma^t \mathcal{R}(s_t, a_t)]$.

In particular, we consider the **continual reinforcement learning** (CRL) setting where the environments are imbued with *added non-stationarity*. While RL agents face inherent non-stationary training dynamics in various forms even in normal MDPs (Igl et al., 2020), we focus on the more challenging problem where the environment itself is non-stationary and the agent needs to adapt.

More concretely, the environment consists of a series of $N$ MDPs $\{\mathcal{M}_0, \mathcal{M}_1, ..., \mathcal{M}_N\}$, where given a training horizon $\mathcal{T}$, the environments switches over to the next MDP every $\tau = \frac{T}{N}$ timesteps.

## 2.2 NETWORK PLASTICITY

We consider feed-forward networks with layers indexed by $\ell = 1, \ldots, L$. Let $W^\ell \in \mathbb{R}^{n_\ell \times n_{\ell-1}}$ and $b^\ell \in \mathbb{R}^{n_\ell}$ denote the parameters that map $h^{\ell-1}$ to pre-activations $z^\ell = W^\ell h^{\ell-1} + b^\ell$ and activations $h^\ell = \sigma(z^\ell)$. By convention, the *incoming* weights for unit $i$ at layer $\ell$ are the $i$-th *row* of $W^\ell$ and the *outgoing* weights are the $i$-th *column* of $W^{\ell+1}$. We use mini-batches $\mathcal{B}_t$ at update step $t$ and write $\mathbb{E}_{x \in \mathcal{B}_t}[\cdot]$ for batch averages.

**Dormant-neuron ratio (Sokar et al., 2023)**  A neuron is *dormant* if its mean activation is small relative to the layer average.

$$\text{DormantRatio}_\ell \;=\; \frac{1}{n_\ell} \sum_{i=1}^{n_\ell} \mathbf{1}\left[ \frac{\mathbb{E}_{x \sim \mathcal{B}_t}|h_i^\ell(x)|}{\frac{1}{n_\ell} \sum_{k=1}^{n_\ell} \mathbb{E}_{x \sim \mathcal{B}_t}|h_k^\ell(x)|} < \tau \right]. \tag{1}$$

where $\tau = 0.1$ in experiments.

**Linearized-neuron ratio (Lyle et al., 2024b)**  We define the number of *linearized units* per layer as the percentage of neurons for which the pre-activation value of a given neuron was positive. We can define linearization as:

$$\text{LinearizedRatio}_\ell = \frac{1}{n_\ell} \sum_{i=1}^{n_\ell} \mathbf{1}\left[ \mathbb{E}_{x \in \mathcal{B}_t}[\mathbf{1}[z_i^\ell(x) > 0]] > \theta \right] \tag{2}$$

where $\theta = 0.9$ in experiments.

These metrics complement each other: Equation 1 captures under-used channels, while Equation 2 captures loss of gating diversity that reduces rank and gradient flow. We report both alongside accuracy/return. Note these definitions are for rectifier based activation functions only. For non-rectifier activations, an analogous variance-based criterion can be used (Lyle et al., 2024a).

## 2.3 UNIT UTILITY AND SELECTIVE RESET

A *neuron-utility score* $S_i^\ell(t)$ quantifies how much unit $i$ at layer $\ell$ contributes to learning and can be computed given network weights $W^\ell$ and activations $h^\ell$ or averaged gradients $\mathbb{E}_{x \in \mathcal{B}_t}[\nabla_{W_i^\ell} \mathcal{L}(W^\ell)]$ where $\mathcal{L}$ is a loss function. To stabilize selection under non-stationarity, it is useful to compute a layer-normalized exponential moving average (EMA) of the score $u_i^\ell(t)$:

$$u_i^\ell(t) \;\leftarrow\; (1-\beta)u_i^\ell(t-1) + \beta \, \tilde{S}_i^\ell(t), \qquad \tilde{S}_i^\ell(t) \;=\; \frac{S_i^\ell(t)}{\frac{1}{n_\ell} \sum_{k=1}^{n_\ell} S_k^\ell(t)}, \quad \beta \in (0,1]. \tag{3}$$

where $\beta \in (0,1]$ is the EMA decay parameter. Normalization over layers (through dividing by the layer mean) makes selection thresholds comparable across layers and training time.

**Selective reset methods**  We study neuron-reset methods that operate immediately after the base optimizer update step and reinitialize units with low utility score $S_i^\ell$ or $u_i^\ell$ based on a *reset rule* $\mathfrak{R}$.

## 3 TRAINING INSTABILITY IN CONTINUAL REINFORCEMENT LEARNING

Training stability is widely recognized as the central challenge in deep reinforcement learning. This brittleness (Sutton et al., 1998; Van Hasselt et al., 2016) arises from the interplay of function approximation and bootstrapping, mechanisms common to both off-policy and on-policy algorithms, and often manifests as instability during training. Even small shifts in the policy can amplify through feedback, spiraling into catastrophic performance collapse (Van Hasselt et al., 2018; Juliani & Ash, 2024). Whilst there can be many sources of instability, Dohare et al. (2023) notably attribute the

phenomenon to sudden, large weight updates that destabilize learning dynamics. Accordingly, numerous major breakthrough in RL have centered on improving stability—through replay buffers in DQN (Mnih et al., 2013), trust regions in TRPO (Schulman et al., 2015a), clipping in PPO (Schulman et al., 2017), and similar innovations.

Continual reinforcement learning exacerbates these stability issues, as the learning objective itself becomes non-stationary. This non-stationarity induces additional challenges, including catastrophic forgetting (French, 1999; Kirkpatrick et al., 2017) and plasticity loss (Dohare et al., 2024; 2021; Liu et al., 2025; Sokar et al., 2023). Neuron-reset–based continual RL methods build on these stabilizing techniques, for example, ReDo over DQN (Sokar et al., 2023; Mnih et al., 2013) and CBP or ReGraMa over PPO and SAC (Dohare et al., 2024; Liu et al., 2025; Schulman et al., 2017; Haarnoja et al., 2018). Yet, by altering network dynamics, such methods can inadvertently reintroduce the very instabilities they aim to overcome.

The aforementioned approaches use a *binary* reset rule $\mathfrak{R}(S_i^\ell, \mathfrak{r}) = \mathbf{1}[S_i^\ell(t) < \mathfrak{r}]^1$ that indicates whether a neuron should be reset if its utility is deemed lower than threshold $\mathfrak{r}$. This then yields the following weight update when the neuron-reset method is applied:

$$W_{i,:}^\ell \leftarrow (1 - \mathfrak{R}(S_i^\ell, \mathfrak{r}))\, W_{i,:}^\ell + \mathfrak{R}(S_i^\ell, \mathfrak{r})\xi_{i,:}^\ell \qquad W_{:,i}^{\ell+1} \leftarrow (1 - \mathfrak{R}(S_i^\ell, \mathfrak{r}))\, W_{:,i}^{\ell+1} \qquad (4)$$

where $\xi_{i,:}^{\ell-1} \sim \mathcal{D}_{\text{init}}$, with $\mathcal{D}_{\text{init}}$ being a weight initialization distribution, for instance Kaiming (He et al., 2015) or Xavier (Glorot & Bengio, 2010). $W_{i,:}^\ell$ denotes the incoming weights into unit $i$ while $W_{:,i}^\ell$ are the outgoing weights.

Notably, under this reset rule, neurons either get reset if they are deemed low-utility, in which case their incoming weights are completely reinitialized, or they do not. This "all or nothing" reinitialization with units getting suddenly reintroduced into the optimization process can in principle lead to gradient spikes which destabilizes training. Additionally, units that are reinitialized have their outgoing weights zeroed, which can lead to a distribution shift in the inputs of the next layer, causing further training instability, especially in the case where layer normalization (Ba et al., 2016) is not used.

We empirically find that these instabilities can and do arise in these algorithms in continual reinforcement learning tasks in Figure 4 and can also lead to policy collapse in Figure 1. Typical policy collapse mitigation such as early stopping (Li et al.) and gradient stabilization approaches such as learning rate annealing (Loshchilov & Hutter, 2016) are designed for stationary, single-task settings and cannot be readily applied to continual learning where ideally the agent should be able to learn throughout its lifetime up to an infinite horizon.

## 4  *Continuous* CONTINUAL BACKPROPAGATION (CCBP)

In order to overcome the aforementioned shortcomings and regain training stability in continual reinforcement learning, we propose *Continuous* Continual Backpropagation (CCBP).

Our method foregoes the use of a binary reset rule and instead computes a smooth, monotonically decreasing map from per-neuron utility scores $u_i^\ell$ to a partial reset coefficient $r_i^\ell$ which is then used to perform *continuous, partial resets* on all network units. This results in incoming unit weights being shifted toward their initialization based on the unit's utility, while the outgoing weights are damped towards zero. CCBP is performed after the base optimizer update every $f$ timesteps, with the exponential moving average of utility metrics being computed every optimization step based on gradient magnitudes. The full algorithm is described in pseudo-code in Algorithm 1.

### 4.1  UTILITY SCORE ESTIMATION

Let $\tilde{S}_i^\ell(t)$ be an instantaneous per-step (or per-mini-batch) utility score. Following recent approaches (Liu et al., 2025; Hernandez-Garcia et al., 2025) we use gradient-based utility criteria, as they have

---

[1] There are slight differences in the reset rule between algorithms. CBP additionally uses a unit's "age" to decide whether it can be reinitialized, while ReDo and ReGraMa only attempt to reset neurons every $n$ timesteps.

been shown to scale more favorably. Concretely, we use the layer-normalized gradient magnitude for a given unit, which is defined by:

$$\tilde{S}_i^\ell = \frac{\mathbb{E}_{x \in \mathcal{B}_t}\left[\left\|\nabla_{W_{i,:}^\ell}\mathcal{L}(x)\right\|\right]}{\frac{1}{n_\ell}\sum_{k=1}^{n_\ell}\mathbb{E}_{x \in \mathcal{B}_t}\left[\left\|\nabla_{W_{k,:}^\ell}\mathcal{L}(x)\right\|\right]} \,. \tag{5}$$

where $\mathcal{L}$ is the loss function being optimized and $h_i^\ell$ is the output of neuron $i$ in layer $\ell$ post-activation.

When computing this utility score, we also adopt the use of running estimates from Dohare et al. (2024) to lower the variance of the estimator. Specifically, we use the exponential moving average of the utility score as defined by Equation 3 and use the resulting $u_i^\ell$ as the utility score rather than $\tilde{S}_i^\ell$.

## 4.2 CONTINUOUS PARTIAL RESETS

The primary driver of increased training stability–and therefore continual learning performance–in CCBP is the use of continuous partial resets. Instead of periodically reinitializing units with low utility, CCBP applies a partial reset transformation to *all* parameters in hidden layers, scaled by their perceived utility.

Let $u_i^\ell$ be a per-layer normalized utility, with mean 1. Instead of defining a reset rule based on a binary threshold such as the one described in section 3, we map utility to a reset intensity $r^\ell \in [0, 1]$ using a function $\phi$:

$$r_i^\ell = \phi(u_i^\ell) = \rho \cdot \min\left(\sigma\left(-\kappa(u_i^\ell - 1)\right), 1\right) \tag{6}$$

where $\sigma$ is a non-linear function and maximum reset fraction $\rho \in (0, 1]$, and sharpness $\kappa > 0$ are hyperparameters, which allow flexible tuning of plasticity based on the applications requirements. We center the inflection point, around the layer mean as deviations from the mean proved sufficient for selection; $\rho$ controls the magnitude of partial resets, and directly adjusts the stability-plasticity tradeoff. Sharpness $\kappa$ can be thought of as the precision parameter, which balances emphasis on low utility neurons vs broader regularization. In its limit $\kappa$ can approach the behavior of the step function $lim_{\kappa \to \infty}[\phi(u)] = \mathbf{1}[\,u < 1\,]$ with $r \to \rho$. With infinite sharpness and a threshold of 1, we would apply partial resets to the least useful half of the network layer each update. With sharpness of zero, this applies partial resets to all neurons equally. However in practice we use a sharpness value in-between, applied to all units increases the regularization effectiveness. In Figure 12 we demonstrate how different $\kappa$ values perform in SlipperyAnt. This plot shows that using the utility function to discriminate between neurons is more effective than a uniform transformation such as in Shrink & perturb.

For the non-linear function of choice, we use the sigmoid function $\sigma(x) = \dfrac{1}{1 + e^{-x}}$. For details on our choice of continuous transformation refer to Appendix C, where we explore alternative options. This function acts to transform the utilities into neuron reset proportions.

The resulting partial reset operator for unit $i$ in layer $\ell$ is

$$W_{i,:}^\ell \leftarrow (1 - r_i^\ell)\,W_{i,:}^\ell + r_i^\ell\,\xi_{i,:}^\ell, \qquad W_{:,i}^{\ell+1} \leftarrow (1 - r_i^\ell)\,W_{:,i}^{\ell+1}. \tag{7}$$

where as in Equation 4, $\xi_{:,i}^{\ell-1} \sim \mathcal{D}_{\text{init}}$. We perform CCBP updates every update frequency $f$ and compute the EMA of utility scores between update steps only. This is implemented by using data from every optimization step to compute the utility score EMA, but along with CCBP updates, we reset utility running averages to their mean 1.

CCBP can be thought of as a targeted decay method, which benefits from the self-regulating accuracy of methods like Shrink & Perturb (Ash & Adams, 2019), whilst learning from the lessons of neuron-reset methods to benefit from precisely targeting less useful neurons.

**Compatibility with optimizers**   Other neuron-reset methods additionally reset underlying optimizer state–such as $\mu$ and $\nu$ parameters in Adam (Kingma & Ba, 2014)–for the affected low-utility units. We did not find base optimizer parameter resets necessary for our method, which reduces overall implementation complexity.

---

**Algorithm 1** CCBP (utilities every step; resets every $f$ steps)

---

**Require:** Update frequency $f$, EMA decay $\beta$, steepness $\kappa$, max reset $\rho$
1: **Input:** current step $t$
2: Compute per-neuron utility $S_i^\ell(t)$                                    ▷ layer-normalized
3: Update EMA: $u_i^\ell \leftarrow (1 - \beta)\, u_i^\ell + \beta\, S_i^\ell$
4: **if** $t \bmod f = 0$ and $t > 0$ **then**                                 ▷ apply resets every $f$ steps
5:     **for** each layer $\ell$ **do**
6:         **for** each neuron $i$ **do**
7:             $r_i^\ell \leftarrow \rho \cdot \min\big( \sigma\big( -\kappa(u_i^\ell - 1)\big),\, 1\big)$              ▷ See Equation 6
8:             $W_{i,:}^\ell \leftarrow (1 - r_i^\ell)\, W_{i,:}^\ell + r_i^\ell\, \xi_{i,:}^\ell$
9:             $W_{:,i}^{\ell+1} \leftarrow (1 - r_i^\ell)\, W_{:,i}^{\ell+1}$
10:            $u_i^\ell \leftarrow 1$                                        ▷ reset running utility after a reset step
11:        **end for**
12:    **end for**
13: **end if**

---

## 5   RESULTS

### 5.1   EXPERIMENT SETUP

We evaluate CCBP against recently proposed continual learning methods in two challenging continual reinforcement learning settings: SlipperyHumanoid and SlipperyAnt. These environments are specialized versions of the Ant and Humanoid environments from OpenAI Gym's MuJoCo environments (Schulman et al., 2015b), similar to the continual RL setting studied by Dohare et al. (2024). SlipperyHumanoid requires a running humanoid to adapt as ground friction changes periodically, while SlipperyAnt[2] requires a running ant-like robot to adapt to friction changes globally. This poses a challenge for continually learning agents, since they must retain locomotion ability while selectively forgetting friction dynamics in their implicit world model.

We train agents for 400 million steps, sample friction log-uniformly from $[0.02, 2.0]$ every 20 million steps and use 2048 parallel environments. To achieve this level of environment parallelism and make experiments of this length tractable, we build these continual RL environments on top of Brax (Freeman et al., 2021), a JAX (Bradbury et al., 2018) based physics engine. We implement our baseline implementations and CCBP in JAX, which allows us to compile the entire continual RL loop end-to-end for increased performance. This enables individual seeds to take less than an hour on a single GPU. We additionally open source the codebase used for these experiments[3].

As the base RL algorithm we use PPO (Schulman et al., 2017) with a rollout size of $196, 608$ and $327, 680$, and for the underlying optimizer we use Adam (Kingma & Ba, 2014) with learning rate $0.0003$ and $0.001$, for SlipperyHumanoid and SlipperyAnt respectively.

In all experiments we evaluate each algorithm with 15 random seeds and report the IQM across seeds, as proposed by Agarwal et al. (2021). Interquartile ranges (IQR) are provided in Appendix F (Table 6). We selected the hyperparameters with highest average final return (Appendix B, Table 3).

---

[2]While SlipperyAnt was studied in prior work (Dohare et al., 2024), implementing it on top of the Brax version leads to different reward and dynamics.

[3]https://anonymous.4open.science/r/continual-learning-82A3/

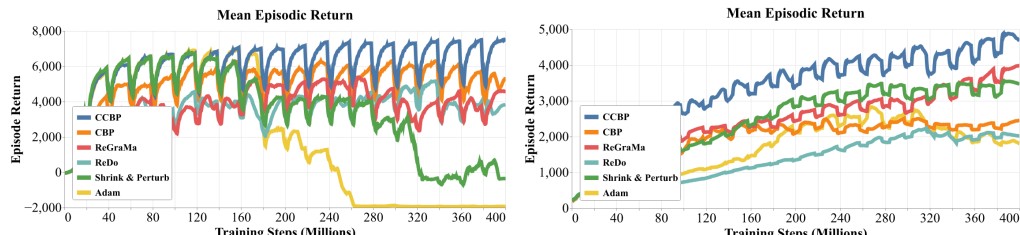

Figure 2: Comparison of IQM of episodic returns over 15 seeds on SlipperyAnt (left) and Slippery­Humanoid (right). IQR is removed for visual clarity and can be found in Appendix F. CCBP maintains high return in SlipperyAnt while leading throughout training in SlipperyHumanoid.

## 5.2 CCBP MAINTAINS OR IMPROVES PEAK PERFORMANCE

On SlipperyAnt, we empirically observe in Figure 2 that CCBP is able to consistently achieve peak performance after each friction change. CCBP maintains this ability even after 400 million steps of training. Vanilla Adam with no neuron-reset method attached achieves strong peak performance but eventually collapses. Meanwhile, other reset-based methods such as ReDo and ReGraMa with a binary reset rule peak later on in training before themselves showing a decline in performance. This illustrates a plasticity-performance trade-off in this setting where existing methods either peak high and collapse, such as Adam which has no mechanism to maintain plasticity, or maintain trainability but are unable to learn optimal policies for each task (friction) due to unstable training dynamics arising from the neuron resets that help maintain gradient flow. In contrast, CCBP is able to achieve both objectives.

Additionally, in SlipperyHumanoid (Figure 2), CCBP is able to significantly outperform baselines in terms of mean episodic return throughout training. Combined with the previous results, this suggests that CCBP might be able transfer learning between tasks more, thus performing continual learning more effectively.

While our primary focus is Reinforcement Learning, we also demonstrate that CCBP mitigates plasticity loss in a continual supervised learning setting (Permuted MNIST). Detailed results showing CCBP outperforming baselines in this regime are provided in Appendix D

## 5.3 CCBP IMPROVES TRAINING STABILITY

In an attempt to elucidate where the performance improvements in CCBP could be coming from, we turn our attention to various network metrics surrounding plasticity and training stability.

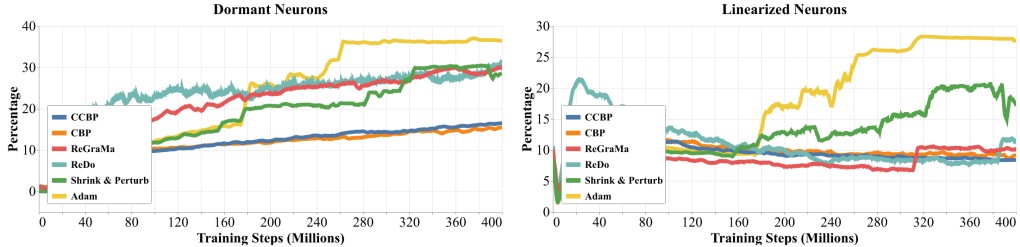

Figure 3: Dormant-neuron ratio (left) and Linearized-neuron ratio (right, lower is better). IQM across 15 seeds. Lower dormancy correlates with higher plasticity and lower linearization correlates with higher representation capacity. CCBP consistently performs among the best alongside CBP.

**Dormant and linearized neurons.** Figure 3 shows that both CCBP and CBP keep the dormant-neuron ratio nearly flat over 400M steps, whereas Adam and baselines that don't use run­ning statistics of the utility score rise steadily, with acceleration after ∼200–300M steps, which coincides with the return degradation in Figure 2. Furthermore, although methods that suppress dor­mancy more strongly tend to have higher linearized-unit ratios initially; CCBP and CBP maintain

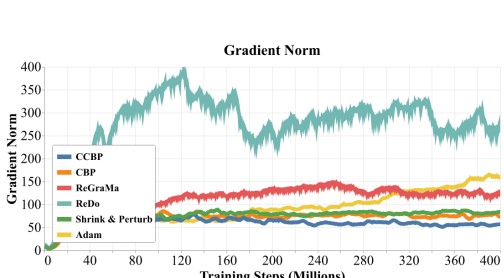

Figure 4: Gradient norm (actor+critic combined, IQM across 15 seeds). Smaller fluctuations correlate with fewer collapses.

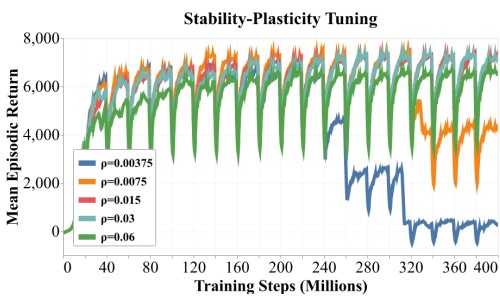

Figure 5: Effect of max per neuron reset $\rho$ on the stability–plasticity tradeoff in SlipperyAnt. Low $\rho$ values preserve stability but hinder long-term adaptation, while high rates enhance plasticity but suppress peak and average performance.

steady dormant and linearized neurons, other methods oscillate in frequency. Overall, it appears that when tuned for performance over prolonged non-stationarity, each algorithm achieves a similar degree of plasticity.

**Gradient norm.** Figure 4 reports combined actor/critic gradient norms. CCBP exhibits smaller fluctuations over time relative to baselines with binary reset rules, which aligns with its ability to avoid collapses and sustain high final return. When compared with the dormant and linearized plots in Figure 3, we can see how increased grad norm correlates with unstable dormant and linearized statistics. These observations are reaffirmed in the SlipperyHumanoid environment, shown in Appendix F (Figure 15).

Overall, this suggests that partial resets do promote training stability.

### 5.4 CCBP PREVENTS POLICY COLLAPSE

Increased training stability appears even more clearly in the unique absence of policy collapse for CCBP among the baselines tested, as shown in Figure 1 and Figure 2.

In SlipperyAnt, we analyze the frequency of policy collapse, which we define as a *drop of 8,000* episodic return from the peak which the agent does not recover from for at least 4 million timesteps, which indicates a significant fall from convergence, and not just a temporary fluctuation caused by periodic dynamics shifts.

### 5.5 CONTROLLING THE STABILITY-PLASTICITY TRADEOFF IN CCBP

In CCBP, the $\rho$ as a direct control over the stability–plasticity balance. A low $\rho$ favors stability, but in challenging environments such as SlipperyAnt, it fails to provide sufficient plasticity for sustained adaptation, leading to eventual instability and performance degradation. Conversely, a high $\rho$ increases plasticity, but at the cost of reduced peak and average performance due to over resetting. This highlights the need to carefully tune $\rho$ to balance adaptability with long-term stability. We empirically demonstrate the effect of ablating this parameter in SlipperyAnt in Figure 5.

## 6 RELATED WORK

**Plasticity Loss in Deep Learning and RL** Deep neural networks under prolonged training on non-stationary data can become harder to adapt (Lyle et al., 2024a; Nauman et al., 2024). This phenomenon, now termed as *loss of plasticity*, has begun to draw increasing attention from the machine learning community. Initially, observations were presented as distinct fields in deep learning, such as class-incremental learning (Savadikar et al., 2023), supervised learning (Ash & Adams, 2020), reinforcement learning (Lyle et al., 2022), and continual learning (Dohare et al., 2021; Nikishin

et al., 2022). However, the seminal work of Dohare et al. (2024) provided a comprehensive study of the phenomenon and unified the community by attributing loss in adaptation performance to rising parameter norms, increased dormant units and reductions in stable rank. These observations motivate approaches that restore favorable weights/feature statistics or re-introduce randomness. Early work turned to L2 regularization (Kumar et al., 2023), or shrink and perturb (Ash & Adams, 2019). These methods globally rescale parameters or add noise to restore trainability during warm-starting and continual training.

**Reinitialization Algorithms**  A prominent line of approaches more closely related to our work attempts to maintain plasticity through directly reinitializing targeted parts of the network, motivated by work that demonstrates how plasticity loss can be attributed to only a subset of neurons which lay dormant (Dohare et al., 2021; Sokar et al., 2023). Work by Hernandez-Garcia et al. (2025) further demonstrates that while simple normalization approaches such as layer norm (Ba et al., 2016) can help, they are not sufficient mitigation and weight reinitialization can further mitigate plasticity loss. Continual Backpropagation (CBP) (Dohare et al., 2021; 2024) identifies inactive neurons by using moving averages of activation statistics and continually resets them. More recent work such as ReDo (Sokar et al., 2023) only initializes dormant neurons periodically and uses activation statistics computed over a batch of single data, simplifying implementation over CBP without cost to performance. Alternative approaches such as SNR (Farias & Jozefiak, 2024) count the time between feature activations and ReGraMa (Liu et al., 2025) motivates gradient magnitudes as more scalable metric for unused neurons than activation statistics.

Our method, Continuous Continual Backpropagation (CCBP), uses gradient magnitude statistics to compute unit utilities similar to ReGraMa, but instead of collecting them over a single batch, a moving average over multiple optimization steps is computed similar to CBP. Additionally, unlike all the aforementioned methods, CCBP does not fully reinitialize a targeted subset of neurons, but instead partially reinitializes all neurons proportional to their utility. Our approach therefore sits between Shrink & Perturb's (Ash & Adams, 2019) global perturbations and full neuron reset methods. Finally, the aforementioned algorithms perform experiments of up to 20 million timesteps, while our experiments have a horizon of 400 million.

# 7 DISCUSSION

In this paper we introduce Continuous Continual backpropagation (CCBP), which applies *partial* resets to neural network parameters, scaled by a utility measure to target low gradient neurons. These periodic partial resets are shown to improve plasticity in two challenging continual reinforcement learning tasks: SlipperyHumanoid, and SlipperyAnt. These reinforcement learning settings introduce considerable non-stationarity through periodic changes to friction values whilst aggressive batch-sizes amplify the issue. This leads inevitably to a collapse in policy performance across baselines. In this setting, CCBP uniquely maintains peak performance throughout. We posit that the partial resets featured in CCBP avoid optimization difficulties that arise in other methods which fully reinitialize neurons based on a binary reset rule and we hypothesize that could contribute to the strong results. Supporting evidence includes observed consistent gradient magnitudes throughout training for CCBP, alongside low dormant and linearized neuron ratio. Finally, we ablate max per neuron reset values in CCBP and showcase how it can be used to trade off plasticity for performance. Overall, partial resets appear to be a promising approach for continual learning at increased training horizons and allows for a wider variety of settings to be tackled than is currently studied in literature.

**Limitations**  Due to computational constraints, our experiments were limited to two continual reinforcement learning environments and one continual supervised learning setting. Future work will study how CCBP interacts with other RL algorithms such as DQN and SAC in continual RL, as well as other network architectures such as ResNets and Transformers in settings such as continual supervised learning. Another promising avenue for future work would be to elucidate theoretically or through empirical ablations the exact mechanism through which hard reset algorithms result in the observed policy collapse.

REPRODUCIBILITY STATEMENT

All of our code, including optimizers, environments, experiments, and plotting scripts, is available through anonymous downloadable source code link provided in the main text (https://anonymous.4open.science/r/continual-learning-82A3/) and will be released open-source under an MIT license. To facilitate reproducibility, said code provides declarative experiment files for generating figures and all project dependencies are specified in the `pyproject.toml` file at the root of the project, with a `uv.lock` file pinning them to specific versions used. The code also provides a comprehensive `README.md` which offers step-by-step instructions for reproducing our results and figures. Finally, we provide full details of hyperparameters used in Appendix B.

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

## A   ROLLOUT SIZE

Increasing rollout size for on-policy RL is known to increase stability as larger batches, rollouts and replay buffers reduce gradient approximation noise, stabilizing training Nikishin et al. (2022). SlipperyAnt already induces gradient noise through periodic dynamic shifts. However, to probe plasticity failures under a realistic compute budget, we adopt a smaller on-policy rollout size. This increases gradient noise and makes collapse observable within a reasonable wall-clock time and more consistent among random seeds. Our conclusions are robust to larger rollouts, as we can still observe significant increases in dormant neurons over time even with drastically larger rollout sizes.

This appendix demonstrates that by increasing rollout-size considerably we are able to mitigate the probability of policy collapse over training time by increasing the batch size by 66%.

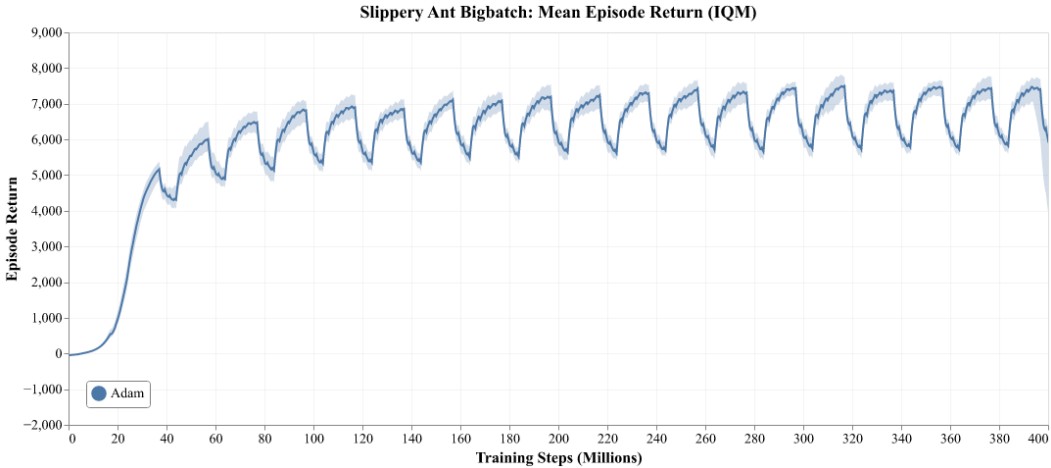

Figure 6: IQM of Adam using larger rollout, with IQR for shaded regions over 15 seeds. This demonstrates large rollouts can delay policy collapse in continual reinforcement learning training

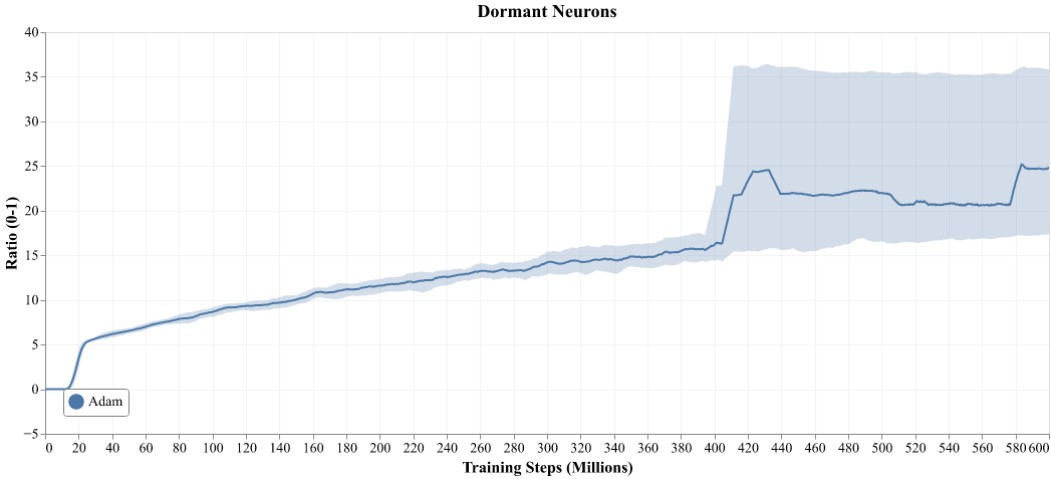

Figure 7: IQM of episodic return using Adam and a larger rollout, with IQR for shaded regions over 15 seeds. Larger batch sizes delay dormant neuron accumulation

Large rollout sizes reduce the variance of gradient estimation. We demonstrate this empirically in Figure 6 where we demonstrate how increasing the rollout size can mitigate the policy collapse problem in the SlipperyAnt environment. However, while large rollout sizes help, they do not remedy the problem of growing parameter norms and dormant neuron accumulation, only dampening their effect. Additionally, large batch sizes can become impractical, requiring more VRAM while being

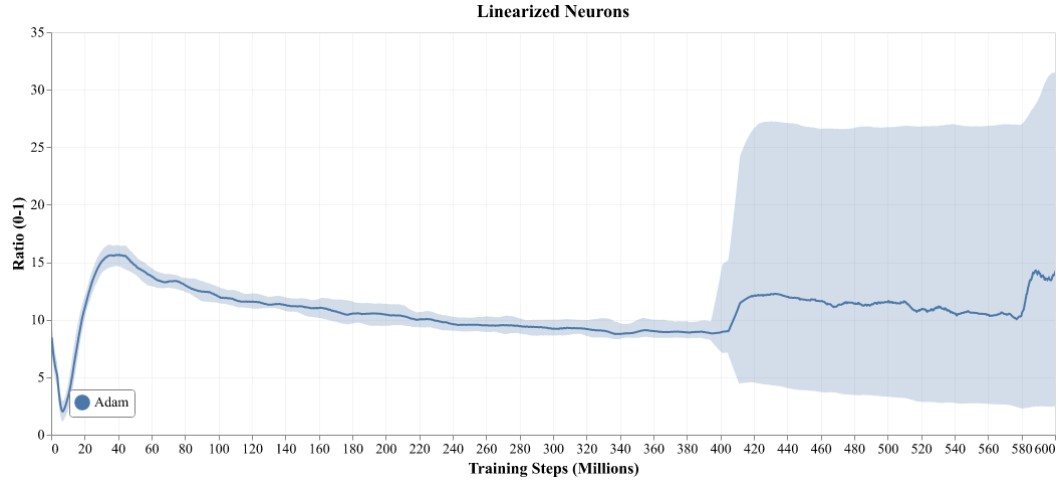

Figure 8: IQM of Linearized neurons using Adam and a larger rollout, with IQR for shaded regions over 15 seeds. Larger batch sizes linearized neuron accumulation

often less sample efficient. This motivates the use of dormant reset methods to directly address neuron dormancy under data non-stationarity.

## B HYPERPARAMETER CONFIGURATION

### B.1 CONTINUAL ANT EXPERIMENT HYPERPARAMETER SETTINGS

The following table summarizes the hyperparameter search space and the optimal values found for each algorithm used in this project.

Table 1: PPO Hyperparameters

| Parameter | Value |
|---|---|
| Learning Rate | 1e-3 (SlipperyAnt), 3e-4 (SlipperyHumanoid) |
| Rollout Steps | 196,608 (SlipperyAnt), 327,680 (SlipperyHumanoid) |
| Number of Epochs | 4 |
| Gradient Steps | 32 |
| Discount Factor ($\gamma$) | 0.97 |
| GAE Lambda ($\lambda$) | 0.95 |
| Entropy Coefficient | 1e-3 |
| Clip Epsilon | 0.2 |
| Value Function Coefficient | 0.5 |
| Normalize Advantages | True |
| Number of Environments | 2048 |
| Number of Tasks | 20 |
| Episode Length | 1000 |
| Steps per Task | 20,000,000 |

## C CONTINUOUS TRANSFORMATION CHOICE

In CCBP we transform utilities such that they map into partial reset fractions. The results of this transformation is multiplied by the max per neuron reset in order to

We consider four candidate transformation functions for mapping utility $u$ into a partial reset factor $\phi(u)$. They are defined as:

Table 2: Network Architecture

| Parameter | Policy Network | Value Function |
|---|---|---|
| Number of Layers | 4 | 5 |
| Hidden Size | 32 | 256 |
| Output Size | 8 | 1 |
| Activation Function | Swish | Swish |
| Kernel Initialization | LeCun Normal | LeCun Normal |
| Data Type | float32 | float32 |

Table 3: Hyperperameters where swept for 5 seeds per configuration, using average final performance as the ranking criteria

| Algorithm | Hyperparameter | Search Space / Values | Optimal Value (SlipperyAnt) | Optimal Value (Humanoid) |
|---|---|---|---|---|
| CCBP | decay_rate | [0.9, 0.99] | 0.99 | 0.99 |
| | max_per_neuron_reset | [0.00, 0.05, 0.10, 0.015, 0.20, 0.25, 0.30] | 0.015 | 0.05 |
| | sharpness | [12, 16, 20] | 16 | 16 |
| | update_frequency | [100, 1000, 10000] | 1000 | |
| CBP | decay_rate | [0.9, 0.99] | 0.99 | 0.99 |
| | replacement_rate | [1e-4, 5e-4, 1e-3, 2.5e-3, 3e-3, 4e-3] | 3e-3 | 2.5e-3 |
| | maturity_threshold | [100, 1000, 10000] | 100 | 100 |
| ReDo | score_threshold | [0.05 to 0.75 (step 0.1; $N$=7)] | 0.65 | 0.5 |
| | max_reset_fraction | [None, 0.05] | None | None |
| | update_frequency | [100, 1000, 10000] | 100 | 100 |
| ReGraMa | score_threshold | [0.05 to 0.75 (step 0.1; $N$=7)] | 0.25 | 0.15 |
| | max_reset_fraction | [None, 0.05] | None | None |
| | update_frequency | [100, 1000, 10000] | 100 | 100 |
| Shrink & Perturb | shrink | [1e-5, 1e-4, 1e-3, 5e-3, 1e-3, 5e-3] | 1e-3 | 1e-3 |
| | perturb | [1e-5, 1e-4, 1e-3, 5e-3, 5e-3, 1e-2] | 5e-3 | 5e-3 |
| | every_n | [100, 1000, 10000] | 1000 | 1000 |

$$\phi_{\text{Exponential}}(u) = \min\left(e^{-\kappa(u-1)}, 1\right)$$

$$\phi_{\text{Sigmoid}}(u) = \min\left(2\,\sigma\left(-\kappa(u-1)\right), 1\right)$$

$$\phi_{\text{Softplus}}(u) = \min\left(\frac{\text{softplus}\left(\kappa(1-u)\right)}{\ln 2}, 1\right) \tag{8}$$

$$\phi_{\text{Linear}}(u) = \max(0, \min(1, 1 - \kappa(u-1)))$$

$$\sigma(z) = \frac{1}{1 + e^{-z}}, \qquad \text{softplus}(z) = \ln\left(1 + e^{z}\right)$$

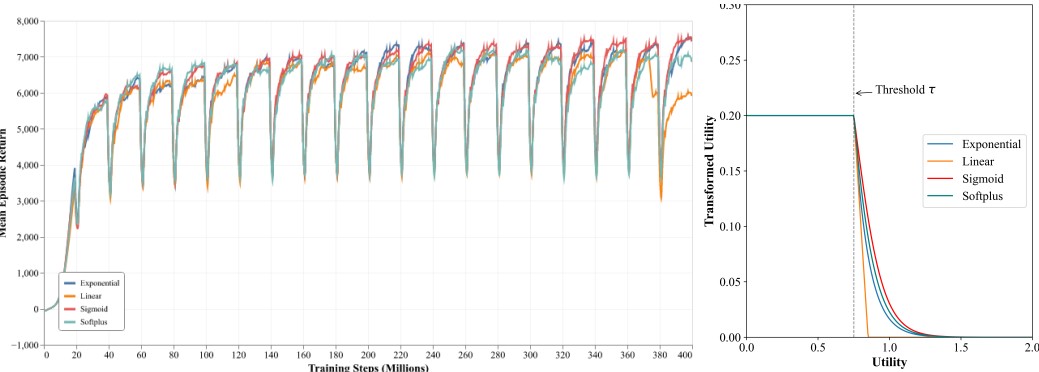

Figure 9: Utility transformation function comparison

| Transformation | Peak | Average | Final |
|---|---|---|---|
| Linear | 8223.515 | 5778.549 | 5873.256 |
| Exponential | 8326.043 | 5999.879 | **7598.191** |
| Softplus | 8145.749 | 5917.624 | 6896.087 |
| Sigmoid | **8360.509** | **6066.286** | 7498.798 |

Table 4: Training performance (IQM). Peak step shown in millions of environment steps.

The choice of transformation function has a subtle but measurable impact on performance. To ensure a fair comparison, we calibrated each function to share the same threshold value. This alignment introduces small differences in the overall area under the curve (AUC), but as shown in Figure 5, these differences are negligible. For instance, although a $\rho$ of 0.015 and 0.03 yield similar performance, their underlying transformation functions differ in AUC by 0.0305, whereas the AUC gap between Linear and Sigmoid transformations in our experiment is only 0.0008. Note, that these transformed utilities are multiplied by $\rho$ in order to get the final partial reset degree. In these transformation function experiments $\rho$ was kept fixed at 0.0015.

The exponential, sigmoid, and softplus functions all amplify resets for neurons with moderately low gradients, leading to improved performance compared to the linear transformation. Sigmoid and exponential behave similarly because their functional forms are close to the mean gradient values where most neurons lie. The small divergence in their behavior for highly dormant neurons appears to have little effect, likely because relatively few neurons fall into that extreme region of the curve.

## D  CONTINUAL LEARNING

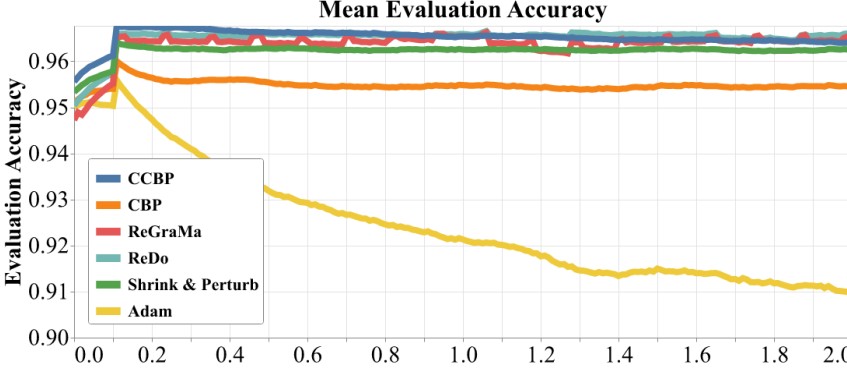

Figure 10: Permuted MNIST Continual Supervised Learning. The Interquartile Mean (IQM) of evaluation accuracy shows CCBP maintaining high plasticity and adaptation efficiency throughout the 2 million step training horizon, preventing the performance degradation observed in standard Adam.

We evaluate CCBP in the Continual Permuted MNIST setting, a supervised learning regime where the input distribution is non-stationary. In this setup, 200 random permutations are applied to the pixels of MNIST characters sequentially throughout the training process.

As shown in Figure 10, CCBP demonstrates the ability to maintain plasticity over long time horizons, effectively matching the stability of reset-based baselines while significantly outperforming standard optimizers like Adam, which exhibit characteristic plasticity loss as the task sequence progresses.

## E  IMPLEMENTATION DETAILS

**Instantaneous and running utility.**    Let $S_i^\ell(t)$ denote the instantaneous utility/score for unit $i$ in layer $\ell$ at update $t$ (defined in Equation 3). We track a running utility via an EMA: Following our reset operator (see below), we reinitialize the running utility to a neutral baseline (we use $u_i^\ell \leftarrow 1$), so that post-reset units compete fairly with mature units over the next estimation window.

**Normalization and gating.** We form a normalized score $\tilde{u}_i^\ell(t)$ within layer $\ell$ to make selection thresholds comparable across layers and time.

**Reset operator.** When resetting a unit $i$ in layer $\ell$:

1. **Incoming weights:** resample the incoming column $W_{:,i}^{\ell-1}$ (and bias $b_\ell[i]$ if present) from the layer's initial distribution (e.g., Kaiming/He).

2. **Outgoing weights:** set the outgoing row $W_{i,:}^\ell$ towards *zero*. This prevents immediate disruption to downstream computations and is the standard reset-method design choice, validated in prior work (Sokar et al., 2023; Dohare et al., 2024) .

**Optimizer and Optax chain.** We provide an easy to use interface for attaching reset methods to optimizer pipelines. Our code is implemented in JAX (Bradbury et al., 2018) with Optax (DeepMind et al., 2020).

A simple Optax chain that matches these choices is:

```
from continual_learning.optim import ccbp
from continual_learning.utils import attach_reset_method
import optax

tx = optax.adam()
tx_w_reset = attach_reset_method(
    ("tx", tx), ("reset_method", ccbp())
)
```

**Note.** The base `tx` can also be composite. In our implementations we set the initial base optimizer to `tx = optax.chain(optax.clip_by_global_norm(0.5), optax.adam())`, as clipping grad norm is standard practice in modern deep learning.

Once attached, the Optax optimizer can be used in the same way as a regular Optax optimizer, only if the reset method requires features (i.e. ReDo or CBP) then these are taken as an input to the `optimizer.update` function.

# F APPENDIX: EXTRA RESULTS

Table 5: SlipperyHumanoid summary: average over training, final return (measured at 400.1M steps for all methods), and peak return with the step at which it occurs. Entries show point estimate with $(+/-)$ IQR. Best final and best peak are in **bold**.

| Method | Average Return | Final Return (at 400M) | Peak Return |
|---|---|---|---|
| CCBP | **3270** $(+488/-465)$ | **4611** $(+425/-443)$ | **6264** $(+647/-1195)$ at 382M steps |
| CBP | 1890 $(+365/-309)$ | 2487 $(+359/-181)$ | 2581 $(+962/-797)$ at 222M steps |
| ReDo | 1340 $(+646/-554)$ | 2021 $(+1189/-1158)$ | 2439 $(+992/-1203)$ at 322M steps |
| ReGraMa | 2393 $(+779/-1180)$ | 4001 $(+431/-1870)$ | 4967 $(+1114/-2690)$ at 382M steps |
| Shrink & Perturb | 2440 $(+652/-671)$ | 3477 $(+603/-1049)$ | 4268 $(+1123/-1567)$ at 282M steps |
| Adam | 1666 $(+994/-838)$ | 1737 $(+1149/-697)$ | 3366 $(+2465/-2390)$ at 262M steps |

CCBP and Adam maintain stable IQR values as they behave the most consistently. Adam will always collapse, whereas CCBP will consistently maintain performance. The variance in baselines, however, arises from the inconsistency in collapse prevention, in other baselines.

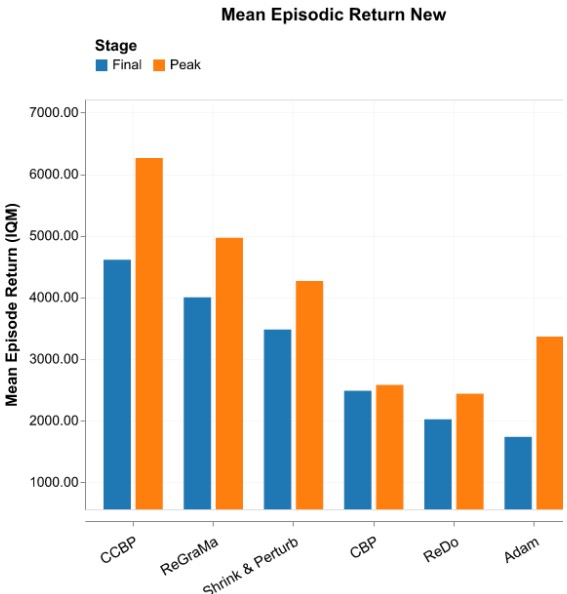

Figure 11: Peak vs. final IQM episodic return in SlipperyHumanoid (higher is better).

Table 6: SlipperyAnt summary: average over training, final return (measured at 400M steps for all methods), and peak return with the step at which it occurs. Entries show point estimate with $(+/-)$ IQR. Best final and best peak are in **bold**.

| Method | IQM Return | Final Return (at 400M) | Peak Return |
|---|---|---|---|
| CCBP | **6051** $(+290/-320)$ | **7494** $(+230/-235)$ | 8380 $(+68/-131)$ |
| CBP | 4932 $(+576/-873)$ | 5643 $(+971/-2723)$ | 7495 $(+426/-787)$ |
| ReDo | 3470 $(+1038/-2434)$ | 3883 $(+1621/-5731)$ | 6459 $(+492/-4437)$ |
| ReGraMa | 3741 $(+1138/-2717)$ | 4635 $(+1387/-6458)$ | 6804 $(+316/-558)$ |
| Shrink & Perturb | 3469 $(+2733/-2902)$ | -341 $(+7820/-1656)$ | 7812 $(+297/-248)$ |
| Adam | 1958 $(+1465/-923)$ | -1921 $(+92/-77)$ | 8034 $(+177/-268)$ |

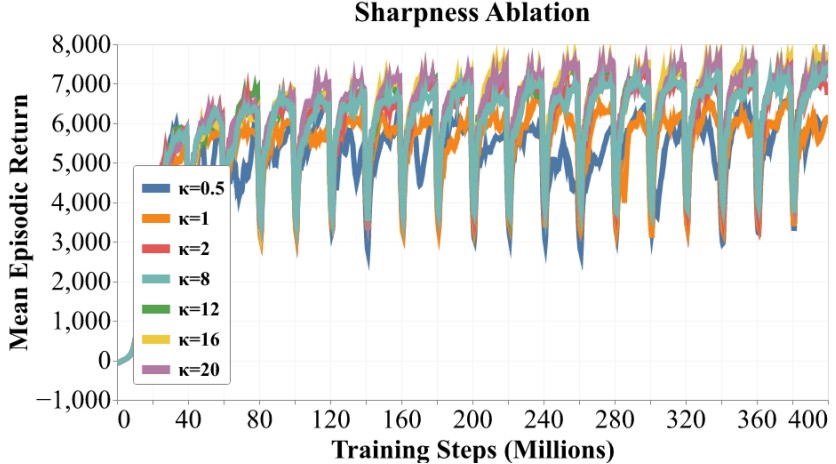

Figure 12: Ablation of the sharpness parameter $(\kappa)$ on the SlipperyAnt environment. Performance is highly consistent across a wide range of values $(\kappa)$, indicating that SCBP is robust to hyperparameter tuning. Notably, low sharpness values $(\kappa \leq 1)$ lead to significant performance degradation, confirming that resets must be targeted at low-utility neurons rather than applied indiscriminately.

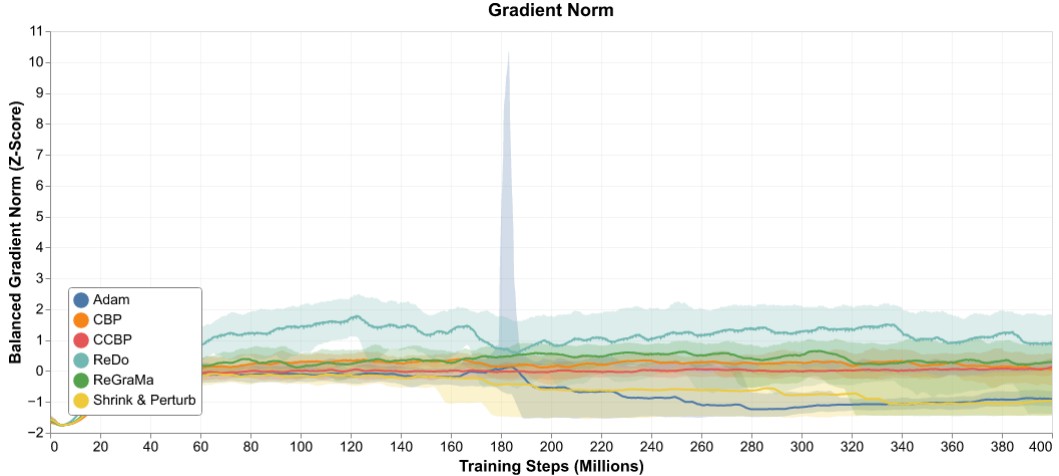

Figure 13: The gradient norm for SlipperyAnt, averaged over both actor and critic with IQM across 15 seeds and IQR as a shaded region. The central spike highlights the point of collapse in Adam, most pronounced as it collapses the most consistently among baselines

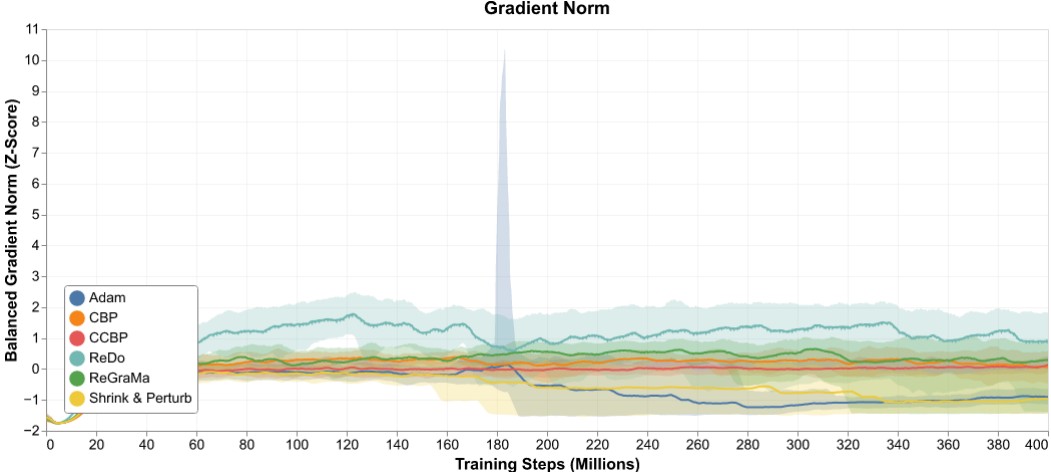

Figure 14: The gradient norm for SlipperyAnt, averaged over both actor and critic with IQM across 15 seeds and IQR as a shaded region. The central spike highlights the point of collapse in Adam, most pronounced as it collapses the most consistently among baselines

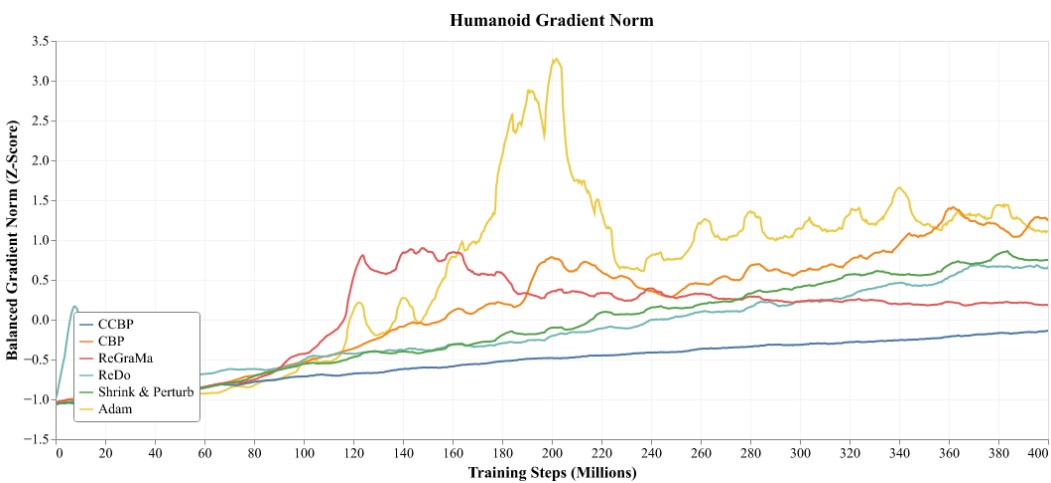

Figure 15: The gradient norm for SlipperyHumanoid, averaged over both actor and critic with IQM across 15 seeds. This demonstrates CCBP maintains stable gradient norm throughout training

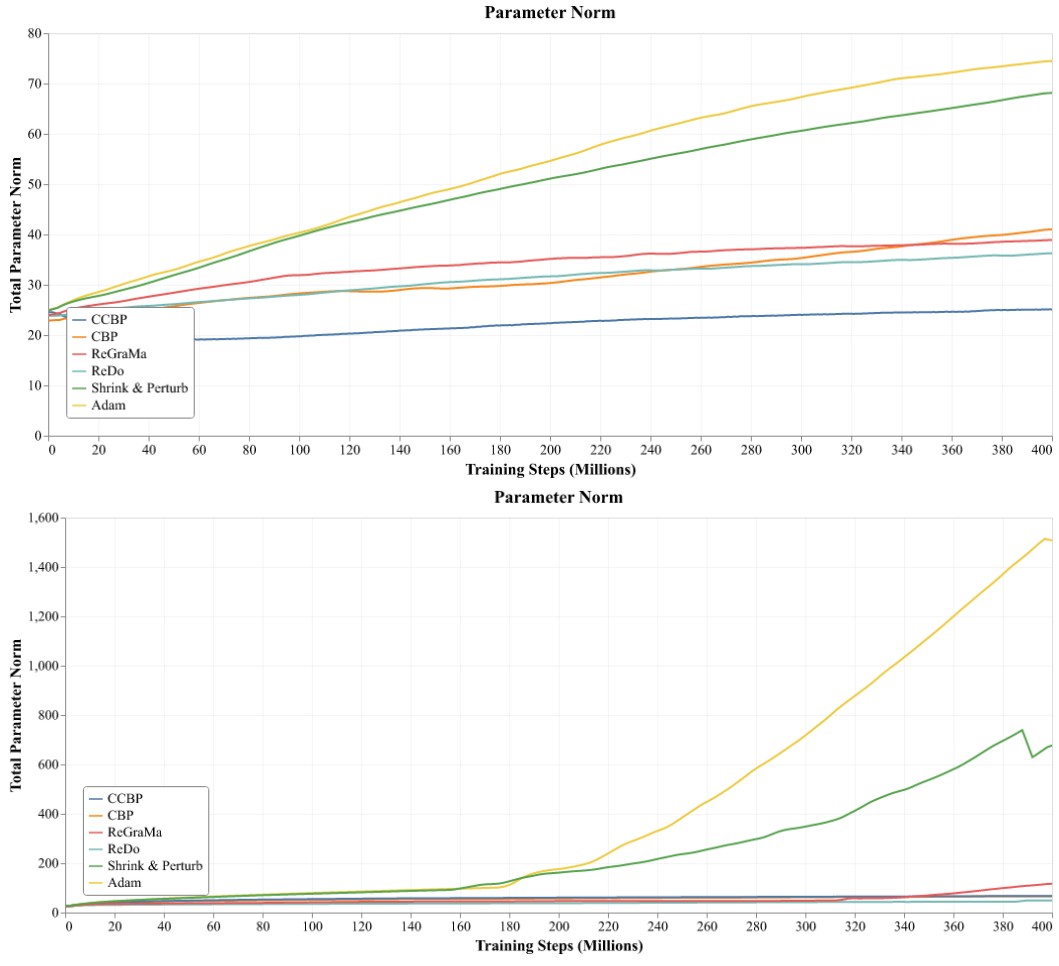

Figure 16: The parameter norm for (top) SlipperyHumanoid and (bottom) SlipperyAnt, averaged over both actor and critic with IQM across 15 seeds. This demonstrates that reset methods are effective at preventing parameter norm growth by targeting high weight valued neurons

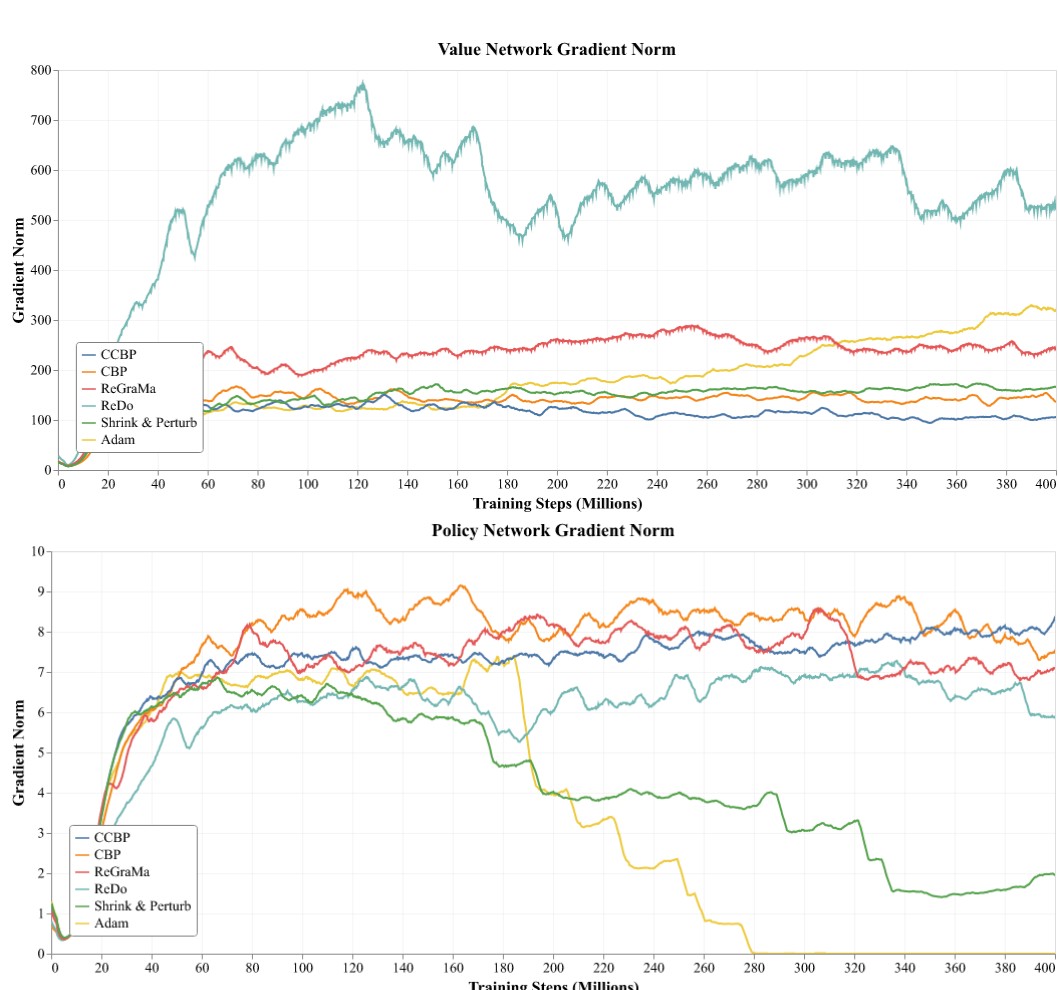

Figure 17: The individual network plots for the (top) Value network and (bottom) Policy network, averaged with IQM across 15 seeds.

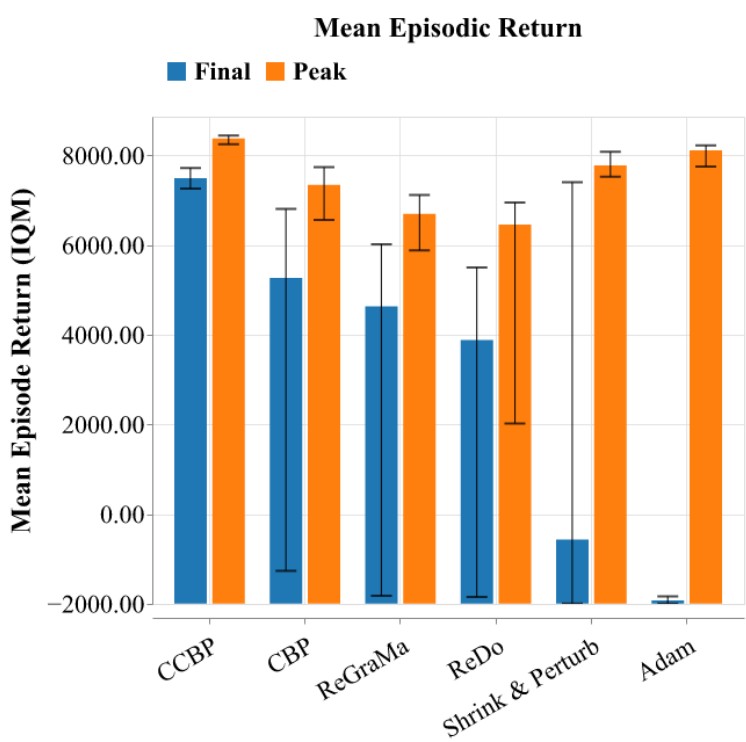

Figure 18: Peak vs. final IQM episodic return in SlipperyAnt. Error bars indicate the IQR across 15 seeds. This figure provides the uncertainty quantification omitted from Figure 1a for visual clarity.

