# OpenReview forum: "Learning Continually at Peak Performance with Continuous Continual Backpropagation"
_ICLR.cc/2026/Conference — Submitted to ICLR 2026_

### Official Review · Reviewer_u1V2 · 2025-11-01

**Soundness:** 2
**Presentation:** 2
**Contribution:** 2
**Rating:** 4
**Confidence:** 4

**Summary:**

This work proposes Continuous Continual Backpropagation (CCBP), an extension of Continual Backpropagation (CBP), to maintain neural network plasticity in continual reinforcement learning. Unlike CBP which uses binary thresholding to identify and reinitialize dormant neurons, CCBP employs a continuous reset mechanism for network units, aiming to stabilize learning in continual learning setting.

**Strengths:**

- CCBP demonstrates improved performance over baselines in two continual RL tasks.
- The paper includes a range of ablation and analysis experiments that help validate the effectiveness of the proposed method.

**Weaknesses:**

- Figure 1 lacks error bars, making it difficult to assess the statistical significance of performance differences.
- The modification from CBP to CCBP appears incremental, with limited conceptual novelty.
- The definition of dormant neurons in Section 2.2 is problematic. For instance, under Tanh activation, a neuron that consistently outputs +1 (saturated but active) would have a high mean activation relative to the layer average and thus not be classified as dormant, even though it contributes no useful gradient signal.
- Experiments are restricted to only two continual RL environments, limiting the generalizability of the claims.
- CCBP introduces two additional hyperparameters, increasing tuning complexity.
- Line 7 in Algorithm 1 should be placed closer to Equation 6 for improved readability and logical flow.

**Questions:**

- How to compute the neuron-utility score $S_i^l(t)$?

---

> ### Author Response · Authors · 2025-11-25
>
> We thank reviewer u1V2 for their time reviewing our work and their feedback on improving clarity in our manuscript and figures.
>
> > Figure 1 lacks error bars, making it difficult to assess the statistical significance of performance differences.
>
> We omitted confidence intervals in Figure 1a because the phenomenon of policy collapse (where the return, the value on the y axis of the plot, drops it -2000) inherently creates massive variance that obscures the plot. As for FIgure 1b, it simply visualises the collapse frequency (the percentage of seeds that collapsed) and thus does not allow for error bars. We provide full uncertainty metrics (IQR) in Table 5 & 6 and have added a version of Figure 1a in Appendix F, Figure 18 with confidence intervals.
>
> > The modification from CBP to CCBP appears incremental, with limited conceptual novelty.
>
> While we understand that CCBP might appear to be an increment over CBP at first glance, especially given its name, we would like to highlight that there are key algorithmic differences which lead to distinct training behaviour and characteristics. CCBP utilises continuous partial resets over CBP's binary hard resets, which overcomes the latter's training instability and collapse. Specifically, CCBP is the only reset-based method we found to maintain performance after 400M Timesteps in SlipperyAnt without collapse. Additionally, CCBP uses different statistics to determine which neurons to reset and by how much compared to CBP. CCBP uses gradient-based statistics to determine neuron utility, while CBP uses activation-based statistics. Therefore, CCBP ends up using two different mechanisms than CBP for its two key algorithmic components, making it qualitatively different to a simple increment, which is reflected in our results.
>
> > The definition of dormant neurons in Section 2.2 is problematic. For instance, under Tanh activation, a neuron that consistently outputs +1 (saturated but active) would have a high mean activation relative to the layer average and thus not be classified as dormant, even though it contributes no useful gradient signal.
>
> We are using the definition introduced by [1], which acknowledges the limitations in its applicability to other activation functions, specifically smooth saturating ones like tanh. However, in our work, we utilize rectifier activation functions such as Swish and ReLU (as detailed in Appendix Table 2), for which this definition is applicable. We have added a note in Section 2.2 to clarify the assumptions this dormant neuron definition makes on the activation function characteristics.
>
> > Experiments are restricted to only two continual RL environments, limiting the generalizability of the claims.
>
>
> We have added evaluation on Continual Permuted MNIST to our revised manuscript (Appendix D), which assess CCBP's performance on Continual Supervised Learning. We find that CCBP performs comparably in this setting as well.
>
>
> > CCBP introduces two additional hyperparameters, increasing tuning complexity.
>
> In the revised version of our manuscript, we have removed the threshold $\tau$ hyperparameter after ablations found it redundant, and have performed further ablations on the sharpness parameter (Appendix F), which demonstrate that it is rather insensitive and does not introduce significant tuning overhead. As a result, only replacement rate, which increases or decreases plasticity, remains as a parameter that is sensitive to tuning on different settings. We hope that eliminating unnecessary hyperparameters and providing greater insight as to the sensitivity of the remaining parameters aids in making CCBP easier to use.
>
> > Line 7 in Algorithm 1 should be placed closer to Equation 6 for improved readability and logical flow.
>
> We have updated our manuscript to revise Equation 6 for consistency with Line 7 in Algorithm 1 in the revised version of our manuscript. Specifically, it now includes the replacement rate parameter to match the algorithm. We hope this addresses this concern. Or is the suggestion to move Algorithm 1 on the same page as Equation 6?
>
> > How to compute the neuron-utility score $S_i^l(t)$?
>
> The neuron utility score varies between our baselines. Our proposed algorithm, CCBP, specifically uses a score based on the gradients of the unit and the layer it resides in, according to Equation 5. We have also updated section 2.3 of our manuscript where the concept is first introduced for additional clarity.
>
> [1] "The dormant neuron phenomenon in deep reinforcement learning." Sokar, Ghada, et al. International Conference on Machine Learning. PMLR, 2023.

---

### Official Review · Reviewer_iU14 · 2025-11-02

**Soundness:** 3
**Presentation:** 2
**Contribution:** 2
**Rating:** 2
**Confidence:** 4

**Summary:**

The paper proposes Continuous Continual Backpropagation (CCBP), a soft-reset variant of Continual Backpropagation (CBP) aimed at mitigating loss of plasticity and policy collapse in continual reinforcement learning. Instead of hard reinitializations (resetting incoming weights to a unit and outgoing weights to zero), CCBP partially pulls the incoming weights toward samples from the initialization distribution while decaying outgoing weights toward zero, using a utility-based modulation derived from per-unit gradient magnitudes. The authors evaluate CCBP on two non-stationary MuJoCo-based environments (SlipperyAnt and SlipperyHumanoid), demonstrating improved performance compared to reset-based baselines such as CBP, ReDo, and ReGraMa.

**Strengths:**

- The paper addresses loss of plasticity, which is an important problem in continual learning.
- The paper’s idea of softening unit reinitialization to improve stability is conceptually simple and intuitively motivated, extending prior work on reset-based plasticity maintenance.
- Clear motivation to soften binary resets for improved performance and sustained plasticity.
- Strong performance on the two tasks discussed (SlipperyAnt and SlipperyHumanoid).

**Weaknesses:**

- The paper's writing requires more work to improve its clarity and accuracy.
- Limited evaluation. The authors only considered two MuJoCo tasks with a single algorithm (PPO)
- Limited ablation. The paper only discusses the sensitivity of the replacement rate parameter. There are many components in the algorithms for which the authors didn’t provide any ablation. For example, why is using Eq. 6 important? Can't we simply use CBP and perform soft resets instead for the units to be replaced, based on CBP statistics? The paper fails to give answers to such important questions by showing a thorough analysis.
- I think the definition of the utility function may be incomplete or incorrect. It was unclear how the authors compute the derivative of the loss wrt the activation $h_i^l$, so I had a quick look at the code, and it seems like the authors do some sort of averaging over the absolute weight gradient, which is not equivalent to what is written in Eq. 5. I might be wrong, so please correct me.


**Less critical issues:**
- The use of the word continuous is confusing in both the title and the paper. Having 'continuous' and 'continual' in the same name is confusing for many readers. I suggest the authors use something along the lines of “Soft Continual Backpropagation” to represent their approach, which is a soft reset version of CBP.
- Linearized-neuron ratio and Dormant-neuron ratio definitions are presented in the background, but no references to the original papers are provided. Is the Linearized-neuron ratio a metric introduced in this work or prior works?
- The hypothesis that hard reset-based algorithms incur instability due to hard resets didn’t hold for CBP, but the authors didn’t discuss it.
- The authors talk about asymptotic performance in the abstract. The asymptotic behavior of a function is not defined if the function oscillates indefinitely, which is the case for the performance plots that continue to oscillate with task changes. I suggest the authors drop using this technical term and replace it with something more suitable. For example, the authors may choose to present the average task performance (averaged over the entire task) on the y-axis, allowing them to discuss asymptotic performance.

**Questions:**

- How well does it work with a single environment?
- The authors didn’t consider other settings where CCBP might be useful, such as with high replay ratios or with other algorithms/networks/action-space types/etc. How well do the authors expect CCBP to perform?
- In line 038, “observe that an increasing number of neurons lay dormant and do not activate throughout training under these conditions” – what conditions?
- In the abstract: “CCBP preserves the long- term performance of standard optimizers” what does that mean exactly? Are the authors comparing CCBP with Adam in stationary setting? The sentence is vague.
- In line 243, “full partial reset” is an oxymoron. I think the authors need to drop the word full?
- “Let $u_l$ i be a per-layer normalized utility, with mean 1” — I don’t see why the mean is 1. Can the authors explain why?
In line 306, “rollout size of 2048 ×32 ×5” — what do 32 and 5 represent here? which one is per-enviornment rollout length?

---

> ### Author Response · Authors · 2025-11-25
>
> We thank reviewer iU14 for thoroughly reviewing our work and appreciate the detailed suggestions on how to strengthen the manuscript’s writing for improved clarity.
>
> > Limited evaluation. The authors only considered two MuJoCo tasks with a single algorithm (PPO)
>
> We focused on PPO as it represents the standard for continuous control and encompasses the core components of most Deep RL algorithms (actor-critic architecture, value function learning). However, to address the concern regarding algorithmic generality, we have added the Continual Permuted MNIST experiments which demonstrate efficacy outside of PPO.
>
> > Limited ablation. The paper only discusses the sensitivity of the replacement rate parameter. There are many components in the algorithms for which the authors didn’t provide any ablation. For example, why is using Eq. 6 important? Can't we simply use CBP and perform soft resets instead for the units to be replaced, based on CBP statistics? The paper fails to give answers to such important questions by showing a thorough analysis.
>
> We have ablated the value of $\tau$ and found that it could be removed, since a threshold value of 1 performs competitively to other values, and have also completed an ablation of the sharpness parameter which we have added in Appendix E. Additionally, whilst we don’t provide direct evaluation of alternative CBP-style activation based utility functions for CCBP, this is the primary characteristic difference between our baselines ReDo and ReGraMa. Finally our sharpness ablation (Appendix F, Figure 12) demonstrates the effectiveness of our utility function. This plot shows as utility function flattens performance decreases, indicating we are targeting unused neurons correctly.
>
> > I think the definition of the utility function may be incomplete or incorrect. It was unclear how the authors compute the derivative of the loss wrt the activation $h_i^l$, so I had a quick look at the code, and it seems like the authors do some sort of averaging over the absolute weight gradient, which is not equivalent to what is written in Eq. 5. I might be wrong, so please correct me.
>
> Our implementation indeed utilizes the mean absolute gradients of the incoming weights rather than the activation gradients. We have updated Equation 5 and the associated text to reflect the use of weight gradients ($\nabla_{W}$).
>
> > The use of the word continuous is confusing in both the title and the paper. Having 'continuous' and 'continual' in the same name is confusing for many readers. I suggest the authors use something along the lines of “Soft Continual Backpropagation” to represent their approach, which is a soft reset version of CBP.
>
> After reflection we agree that Continuous, while not an inappropriate description of our algorithm, can be a suboptimal word choice when put next to Continual. We plan to adopt the reviewer's suggestion and rename the method to Soft Continual Backpropagation (SCBP) to better reflect its relationship to CBP and Soft Resets in the camera ready manuscript.
>
> > Linearized-neuron ratio and Dormant-neuron ratio definitions are presented in the background, but no references to the original papers are provided. Is the Linearized-neuron ratio a metric introduced in this work or prior works?
>
> We thank the reviewer for noticing the missing citation, we have added the appropriate citations to both dormant and linearized neuron metric definitions for clarity; Linearized neurons were introduced by Lyle et al., 2024 [1].
>
> > The hypothesis that hard reset-based algorithms incur instability due to hard resets didn’t hold for CBP, but the authors didn’t discuss it.
>
> We note that while CBP overall performs well in our results relative to other baselines, it still collapses in 3 of our 15 test seeds as can be seen in Figure 1b. Nevertheless, we agree that further work elucidating empirically or theoretically the mechanism through which hard reset algorithms result in policy collapse is needed, but we deem that outside the scope of our paper and leave it to future work. We have updated our manuscript to reflect this in our Limitations section.
>
>
> [1] "Disentangling the causes of plasticity loss in neural networks." Lyle, Clare, et al. arXiv preprint arXiv:2402.18762 (2024).

---

> ### Author Response · Authors · 2025-11-25
>
> > The authors talk about asymptotic performance in the abstract. The asymptotic behavior of a function is not defined if the function oscillates indefinitely, which is the case for the performance plots that continue to oscillate with task changes. I suggest the authors drop using this technical term and replace it with something more suitable. For example, the authors may choose to present the average task performance (averaged over the entire task) on the y-axis, allowing them to discuss asymptotic performance.
>
> We thank the reviewer for their insight into the correct use of terminology in the continual learning setting. We have revised our use of terminology to avoid the term "asymptotic performance" in the revised manuscript in favor of language more accurately descriptive of the algorithm's capacity to recover and retain high rewards after environmental shifts.
>
> > How well does it work with a single environment?
>
> We would like to ask for further clarification. Is single environment here meant to refer to not using environment parallelism / vectorised environments during training, or is it meant to refer to studying single-task reinforcement learning as a continual learning setting?
>
>
> > The authors didn’t consider other settings where CCBP might be useful, such as with high replay ratios or with other algorithms/networks/action-space types/etc. How well do the authors expect CCBP to perform?
>
> We have added evaluation on Continual Permuted MNIST to our revised manuscript, which demonstrates that CCBP still performs in continual learning settings such as continual supervised learning which are distant to our existing continual RL environments
>
> > In line 038, “observe that an increasing number of neurons lay dormant and do not activate throughout training under these conditions” – what conditions?
>
> The conditions the text was referring to were the constant shifts in training distribution. We have revised our manuscript as follows for further clarity: "we observe that shifts in the training distribution cause an increase in previously active neurons becoming dormant."
>
> > In the abstract: “CCBP preserves the long- term performance of standard optimizers” what does that mean exactly? Are the authors comparing CCBP with Adam in a stationary setting? The sentence is vague.
>
> We agree that the sentence in question is vague. For clarity, we are not comparing performance of CCBP and Adam in the stationary setting. We have updated the phrasing in the abstract of our revised manuscript accordingly: "CCBP outperforms both decay-based and reset-based methods after long sequences of distribution shifts, and uniquely prevents policy collapse in challenging continual reinforcement learning environments”.
> > In line 243, “full partial reset” is an oxymoron. I think the authors need to drop the word full?
> We agree that this wording was inadequate. This phrase was related to the description of threshold parameter $\tau$ which we have since ablated and found to be redundant, and have made the decision to remove it as a result. This confusing phrase has also been removed from our manuscript as it is no longer relevant.
>
> > “Let $u_l$ i be a per-layer normalized utility, with mean 1” — I don’t see why the mean is 1. Can the authors explain why?
>
> The utility, calculated in Equation 5, is a moving average of scores $S_i^\ell(t)$. These scores are defined for each neuron and divided by the mean in the right hand side’s denominator. This division by the average results in centering the utility distribution around 1.
>
> > In line 306, “rollout size of 2048 ×32 ×5” — what do 32 and 5 represent here? which one is per-environment rollout length?
>
> Presenting the rollout size as a product of these three factors was a result of how the rollout phase of PPO is parameterised in our implementation, with 2048 environments and 32 x 5 = 160 per-environment rollout lengths. To avoid confusion, we have replaced it with the integer it represents: 327680.

---

> > ### Comment · Reviewer_h1wL · 2025-11-27
> >
> > I thank the authors for making revisions based on the feedback. Removing the threshold hyperparameter makes the algorithm a lot easier to use and tune. The new experiments in supervised learning and new baselines make the paper stronger. I have updated my score based on these changes.
> >
> > However, I think the main issue with the paper is still that the evaluation of CCBP is limited. I suggest testing it in stationary RL environments like "Dog Walk" and "Humanoid Stand" as done by Liu et al. (2025).
> >
> > The paper needs to be polished. The authors mention that the threshold hyperparameter is now removed. However, the hyperparameter is still mentioned on line 280, as well as in hyperparameter tables, among other places.
> >
> > Liu et al. (2025) Measure gradients, not activations! Enhancing neuronal activity in deep reinforcement learning.

---

> ### Author Response · Authors · 2025-12-03
>
> We thank the reviewer for their feedback, we have fixed the formatting issues with the manuscript such that the threshold parameter does not occur in text. Whilst we agree that results on single task RL would be of interest, they’re outside the scope of this work, which focuses specifically on more challenging continual learning settings with added non stationarity stemming from task changes.

---

### Official Review · Reviewer_h1wL · 2025-11-03

**Soundness:** 3
**Presentation:** 3
**Contribution:** 3
**Rating:** 4
**Confidence:** 4

**Summary:**

This paper proposes a new algorithm for continual learning, called Continuous Continual Backpropagation (CCBP). CCBP resets low utility neurons in the network in a softer manner than previous resetting methods. The paper claims that smoother resets lead to more stable learning, which improves performance. The method is evaluated on two non-stationary RL environments. In both environments, CCBP outperforms all prior neuron-reinitialization methods.

**Strengths:**

* Empirical analysis is well done. A wide range of settings for hyperparameters is tested for all algorithms, and all algorithms are evaluated on 15 seeds.
* CCBP is a good way to extend prior neuron-reinitialization methods. The specific proposal for smoothing resets allows for fine-grained control on how exactly neurons should be reset.
* CCBP performs well in both of the tested environments. The performance of CCBP is particularly impressive in Slippery Humanoid.

**Weaknesses:**

Limited environments: The proposed algorithm is only evaluated on two environments. That is not sufficient. The algorithm should be tested on other environments. Non-stationary RL environments, such as the sequential Atari environments proposed by Abbas et al. (2023), or continual supervised learning problems like class-incremental CIFAR, and even simpler problems like random label MNIST, could be useful for validating the effectiveness of CCBP and its sensitivity to its hyperparameters.

Extra hyperparameters:
* The algorithm adds five new hyperparameters. Two of these hyperparameters, "decay-rate" and "update-frequency," are fairly intuitive and easy to set in most applications. But for the remaining three hyperparameters, there is no clear intuition on how to set them.
* The authors already conducted a wide grid search for CCBP's hyperparameters on the two non-stationary RL environments. The results for those experiments can reveal the sensitivity of CCBP to these hyperparameters. Further evaluation of CCBP in other environments will further reveal the sensitivity of CCBP to its hyperparameters.
* Another suggestion is to plot Equation 6, with utility ($u_i$) on the x-axis and the fraction of reset ($r_i$) on the y-axis, for different values of $\kappa$, $\tau$, and $\rho$. These plots will help the reader understand CCBP.
* And perhaps most importantly for hyperparameters, Table 3 in the appendix shows that the highest value of threshold tested was 0.95, and it was the best-performing value in both environments. This raises the obvious question: What if the threshold, $\tau$, was 1? Would it perform better or similar to 0.95? If it does, then the default value of $\tau$ can be set to 1. That eliminates both hyperparameters $\kappa$ and $\tau$, and essentially Equation 6, making CCBP much simpler and easier to use.

**Questions:**

1. What is the unit on the y-axis of Figure 1b? What does "collapse frequency" mean? I think it's the number of seeds (out of 15) in which the algorithm collapsed. But please make it clear in the figure exactly what the y-axis is.
2. Where is evidence for the first contribution? The first claimed contribution is "We demonstrate that current reset-based continual learning algorithms suffer from unstable training dynamics." The closest thing to evidence is Figure 4. But Figure 4 merely shows that the gradient norm is lower in CCBP. Higher gradient norm does not mean "unstable training dynamics". Something like "Largest total weight change in the network," as plotted by Dohare et al. (2024) in Extended Data Fig. 5c, is a more direct evidence of unstable training dynamics. However, even that does not directly mean unstable training dynamics. Please either remove this claimed contribution or provide more direct evidence of "unstable training dynamics."
3. In Equation 5, is $h_i$ the post-activation or the pre-activation value?
4. I suggest writing line 7 of Algorithm 1 alongside Equation 6. Currently, Equation 6 does not fully represent the relationship between utility and the amount of reset, as well as its connection to all the hyperparameters.
5. The name "replacement-rate" for hyperparameter $\rho$ seems imprecise. "replacement-rate" as used in CBP has a very specific meaning of fraction of units replaced per step. However, the hyperparameter $\rho$ in CCBP has a very different function. It determines the maximum proportion by which a unit will be reset. I suggest changing the name of this hyperparameter. Something like "max_per_neuron_reset" will be a more precise name.
6. What exactly does rollout size on line 306/307 (rollout size of 2048 × 32 × 5) mean? What exactly are the dimensions of 2048, 32, and 5?
7. Was layer normalization used in these networks? Addition of two baselines, one where learning networks have LN, and the second where these networks include LN with L2 regularization. These two baselines will provide further perspective on CCBPs' performance.

---

> ### Author Response · Authors · 2025-11-25
>
> We thank reviewer h1wL for their careful review of our work and their insightful feedback on simplifying our algorithm and reducing its tuning complexity.
>
> > Limited environments: The proposed algorithm is only evaluated on two environments. That is not sufficient. The algorithm should be tested on other environments. Non-stationary RL environments, such as the sequential Atari environments proposed by Abbas et al. (2023), or continual supervised learning problems like class-incremental CIFAR, and even simpler problems like random label MNIST, could be useful for validating the effectiveness of CCBP and its sensitivity to its hyperparameters.
>
> We have added results for Continual Permuted MNIST to evaluate CCBP in a Continual Supervised Learning context in Appendix D. We chose this setting to strictly test the optimization algorithm's ability to handle distribution shifts without the confounding factors of RL exploration. Regarding Atari, we prioritized the Brax environments (Ant/Humanoid) because they are computationally efficient. This was necessary to rigorously measure policy collapse over long continual learning training runs and for 15 seeds.
> > The algorithm adds five new hyperparameters. Two of these hyperparameters, "decay-rate" and "update-frequency," are fairly intuitive and easy to set in most applications. But for the remaining three hyperparameters, there is no clear intuition on how to set them.
>
> > The authors already conducted a wide grid search for CCBP's hyperparameters on the two non-stationary RL environments. The results for those experiments can reveal the sensitivity of CCBP to these hyperparameters. Further evaluation of CCBP in other environments will further reveal the sensitivity of CCBP to its hyperparameters.
>
> > Another suggestion is to plot Equation 6, with utility ($u_i$) on the x-axis and the fraction of reset ($r_i$) on the y-axis, for different values of $\kappa$, $\tau$ and $\rho$. These plots will help the reader understand CCBP.
>
> > And perhaps most importantly for hyperparameters, Table 3 in the appendix shows that the highest value of threshold tested was 0.95, and it was the best-performing value in both environments. This raises the obvious question: What if the threshold, $\tau$ , was 1? Would it perform better or similar to 0.95? If it does, then the default value of  can be set to 1. That eliminates both hyperparameters $\kappa$ and $\tau$, essentially Equation 6, making CCBP much simpler and easier to use.
>
> After further ablations, we found that the threshold parameter ($\tau$) is not sensitive; a value of 1.0 performs competitively across settings. Consequently, we have removed $\tau$ entirely, simplifying Equation 6. Furthermore, we include a new ablation (Appendix F, Figure 12) which provides insights into the impact of "sharpness” ($\kappa$). This leaves replacement_rate (now max_per_neuron_reset) as the single primary "dial" for tuning the plasticity-stability trade-off, (which itself is ablated in Figure 5) making CCBP comparable in tuning complexity to existing methods.
>
> > 1. What is the unit on the y-axis of Figure 1b? What does "collapse frequency" mean? I think it's the number of seeds (out of 15) in which the algorithm collapsed. But please make it clear in the figure exactly what the y-axis is.
>
> We have revised the figures description to be: “Right - number of seeds where policy collapses during training, with 15 seeds total (see section 5.4)”
>
> > 2. Where is evidence for the first contribution? The first claimed contribution is "We demonstrate that current reset-based continual learning algorithms suffer from unstable training dynamics." The closest thing to evidence is Figure 4. But Figure 4 merely shows that the gradient norm is lower in CCBP. Higher gradient norm does not mean "unstable training dynamics". Something like "Largest total weight change in the network," as plotted by Dohare et al. (2024) in Extended Data Fig. 5c, is a more direct evidence of unstable training dynamics. However, even that does not directly mean unstable training dynamics. Please either remove this claimed contribution or provide more direct evidence of "unstable training dynamics."
>
> We define 'unstable training dynamics' in the context of RL as sudden, irreversible performance degradation. Figure 1b provides the direct evidence: hard-reset methods (CBP, ReDo) frequently trigger catastrophic policy collapse (return dropping to -2000) following environmental shifts. In contrast, CCBP maintains zero collapses across all seeds. We attribute this to the partial reset mechanism, which avoids the large, instantaneous shifts in parameter space caused by binary re-initialization, allowing the agent to adapt without destroying the existing policy.
>
> > 3. In Equation 5, is $h_i$ the post-activation or the pre-activation value?
>
> $h_i$ is the post-activation value. We have updated the text to clarify this.

---

> ### Author Response · Authors · 2025-11-25
>
> > 4. I suggest writing line 7 of Algorithm 1 alongside Equation 6. Currently, Equation 6 does not fully represent the relationship between utility and the amount of reset, as well as its connection to all the hyperparameters.
>
> We have added the replacement rate/max per neuron reset  $\rho$ to Equation 6 to better reflect how utility maps to reset intensity. This makes it consistent with Line 7 of Algorithm 1. We hope this addresses this concern. Or is the suggestion to move Algorithm 1 on the same page as Equation 6?
>
> > 5. The name "replacement-rate" for hyperparameter $\rho$ seems imprecise. "replacement-rate" as used in CBP has a very specific meaning of fraction of units replaced per step. However, the hyperparameter $\rho$ in CCBP has a very different function. It determines the maximum proportion by which a unit will be reset. I suggest changing the name of this hyperparameter. Something like "max_per_neuron_reset" will be a more precise name.
>
> To avoid confusion, we have renamed the parameter to "max_per_neuron_reset"
>
> > 6. What exactly does rollout size on line 306/307 (rollout size of 2048 × 32 × 5) mean? What exactly are the dimensions of 2048, 32, and 5?
>
> Presenting the rollout size as a product of these three factors was a result of how the rollout phase of PPO is parameterised in our implementation. To avoid confusion, we have replaced it with the integer it represents: 327680.
>
> > 7.  Was layer normalization used in these networks? Addition of two baselines, one where learning networks have LN, and the second where these networks include LN with L2 regularization. These two baselines will provide further perspective on CCBPs' performance.
>
> We appreciate the reviewer's suggestion for additional baselines. At this point we have added Soft Shrink & Perturb as an additional baseline and CCBP still showed stronger performance overall, particularly on humanoid and we are in the process of regenerating plots. We will incorporate these new results into the camera ready manuscript.

---

### Official Review · Reviewer_xF2J · 2025-11-07

**Soundness:** 2
**Presentation:** 3
**Contribution:** 2
**Rating:** 2
**Confidence:** 4

**Summary:**

This paper proposes an algorithm called Continuous Continual Backpropagation. Instead of using the neuron utility score as something used to decide whether the neuron should be reset based on some threshold, as several other works do, this work uses the utility score to do soft partial resets of every neuron based on its utility score. The reset is done similarly to how Shrink and Perturb does it, but with a different strength for each neuron based on its utility. The evaluation of the algorithm is done on Continual RL environments Slippery Humanoid and SlipperyAnt where the friction coefficient int the environment changes for every task.

**Strengths:**

- The method is intuitive, it makes sense that doing partial resets of neurons could be a better way of resetting neurons, rather than thresholding.
- The paper shows that partial resets can help mitigate policy collapse when training in a continual RL setting.

**Weaknesses:**

- I think it would be important to see the performance of this method in other settings as well. This could be something like the continual learning or warmup setting from [1] or maybe even single task RL.
- It’s not clear based on the metrics provided why CCBP is better than say CBP. The metrics of dormant/linearized neurons and of gradient norm are essentially the same, and there isn’t evidence showing that what the paper says is happening is the reason for the improved performance.
- The hyperparameters searched over for Shrink and Perturb seem too conservative compared to what is used in the literature [1,2]. I am not sure it is a fair comparison.
- Eq 6 is missing $\rho$, and it seems to be called r in some places and $\rho$ in others.

**Questions:**

- How does a method like Soft Shrink and Perturb [2], that essentially does SnP on every step at a much smaller strength, do in this setting?
- What does this line mean: “All baselines and CCBP were tuned for the stability–plasticity trade-off”?

---

> ### Author Response · Authors · 2025-11-24
>
> We thank reviewer xF2J for their time reviewing our work. We appreciate their valuable feedback on how to further evaluate our algorithm and demonstrate its strengths.
>
> > I think it would be important to see the performance of this method in other settings as well. This could be something like the continual learning or warmup setting from [1] or maybe even single task RL.
>
> We agree that demonstrating generality is crucial. To address this, we have expanded our evaluation to include Continual Permuted MNIST. This introduces a new test setting (Continual Supervised Learning) with different optimization dynamics. The results (added to Appendix) demonstrate that CCBP maintains plasticity with greater learning efficiency than baselines in this setting as well.
>
> > It’s not clear based on the metrics provided why CCBP is better than say CBP. The metrics of dormant/linearized neurons and of gradient norm are essentially the same, and there isn’t evidence showing that what the paper says is happening is the reason for the improved performance.
>
> While the levels of plasticity (dormant/linearized ratios) are similar between CBP and CCBP, the cost to achieve them differs. CBP uses hard resets, which induce gradient spikes (instability) that can lead to policy collapse. CCBP achieves the same necessary level of plasticity via partial resets, maintaining a lower and more stable gradient norm (as seen in Figure 4). The evidence for this improved trade-off is the difference in Policy Collapse Frequency (Figure 1b), where CCBP is the only method with zero collapses while maintaining high return.
>
> > The hyperparameters searched over for Shrink and Perturb seem too conservative compared to what is used in the literature [1,2]. I am not sure it is a fair comparison.
>
> We found the slightly larger shrink and perturb factors in our paper to be more suitable given the intervals we wanted to test and keep consistent between methods (10, 1000, 10000). We have re-tuned Shrink and Perturb using the hyperparameter ranges suggested in recent literature [1]. However, even with these updated parameters (specifically checking smaller perturbation and shrink magnitudes), we did not observe a performance improvement over the baselines reported in our initial submission. We have updated the paper to reflect these additional tuning efforts. To ensure consistency with standard literature (e.g., Ash & Adams., 2020 [2]), we have also updated our notation in Table 3 to report the shrinkage magnitude (e.g., 1e-2) rather than the multiplier (0.99).
>
> > Eq 6 is missing, $\rho$ and it seems to be called r in some places and  in others.
>
> We have fixed this in the revised version of the manuscript. Eq 6 now includes the replacement rate $\rho$ with alternate notation corrected.
>
> > How does a method like Soft Shrink and Perturb [2], that essentially does SnP on every step at a much smaller strength, do in this setting?
>
> We have implemented and evaluated Soft Shrink and Perturb as an additional baseline. While it performed well in SlipperyAnt, it struggled significantly in the more challenging SlipperyHumanoid environment and Continual Supervised Learning settings. This highlights that CCBP’s targeted approach (scaling resets by utility) offers robustness that untargeted soft perturbations do not. We will add these new results into the camera ready manuscript.
>
> > What does this line mean: “All baselines and CCBP were tuned for the stability–plasticity trade-off”?
>
> We agree this wording is ambiguous and have revised the phrasing of this to “ We selected the hyperparameters with highest average final return” for clarity.
>
> References
>
> [1] "Loss of plasticity in deep continual learning." Dohare, Shibhansh, et al. Nature 632.8026 (2024): 768-774.
>
> [2] "On warm-starting neural network training." Ash, Jordan, and Ryan P. Adams. Advances in neural information processing systems 33 (2020): 3884-3894.

---

### Meta-Review · Area_Chair_Au54 · 2026-01-02

**Summary:**

In their initial review, all four reviewers recommended rejection (with two of those four reviewers only doing so "marginally").

An important concern that in some form is raised by all four reviewers is the limited scope of the empirical evaluation. Specifically, in the initial version of the paper the proposed method was evaluated in only two RL environments and against a somewhat limited set of baselines.
In their rebuttal, the authors respond by adding an additional evaluation of their proposed method (permuted MNIST) and by adding additional baselines. While some reviewers had asked for experiments in a stationary RL experiment, the authors responded that such an experiment is beyond the scope of the current paper.

Other concerns were raised as well, but the limited scope of the empirical evaluation, and the shared concern by all reviewers in this regard, is already enough for me to suggest this paper not to be accepted.

**Reviewer Concerns:**

Only one of the four reviewers responded to the author rebuttal before the discussion period was closed prematurely. This reviewer (h1wL) indicated they thought the added experiments made the empirical evaluation of the paper somewhat stronger, but they also indicated that they were still not fully convinced by the scope of the empirical evaluation.

The other reviewers did not reply to the author rebuttal, but my expectation is that they would also not be fully satisfied by the extent by which the empirical evaluation had been extended.

**Reviewer Scores:**

Reviewer h1wL indicated in their response that they updated their score in response to the author rebuttal. Their initial score was a 4. Given that Reviewer h1wL indicated to still not be fully convinced, I expect they raised their score to at most a 6.

My expectation is that the other reviewers would also not have been fully convinced by the author rebuttal. Perhaps that one or two of the other reviewers might have also slightly increased their score, but I expect that none of the reviewers would have clearly supported acceptance.

---

### Decision · Program_Chairs · 2026-01-26

Reject